# Fira: Can We Achieve Full-rank Training of LLMs under Low-rank Constraint?

**Xi Chen**[1], **Kaituo Feng**[1], **Changsheng Li**[1*], **Xunhao Lai**[2],
**Xiangyu Yue**[3], **Ye Yuan**[1], **Guoren Wang**[1,4]
[1]Beijing Institute of Technology
[2]School of Intelligence Science and Technology, Peking University
[3]MMLab, The Chinese University of Hong Kong
[4]Hebei Province Key Laboratory of Big Data Science and Intelligent Technology
xichen.fy@gmail.com, kaituofeng@gmail.com,
lcs@bit.edu.cn, laixunhao@pku.edu.cn,
xyyue@ie.cuhk.edu.hk, yuan-ye@bit.edu.cn, wanggrbit@126.com
https://github.com/xichen-fy/Fira

## Abstract

Low-rank training has emerged as a promising approach for reducing memory usage in training Large Language Models (LLMs). Previous methods either rely on decomposing weight matrices (e.g., LoRA), or seek to decompose gradient matrices (e.g., GaLore) to ensure reduced memory consumption. However, both of them constrain the training in a low-rank subspace, thus inevitably leading to sub-optimal performance. To resolve this, we propose a new plug-and-play training framework for LLMs called Fira, as the first attempt to consistently preserve the low-rank constraint for memory efficiency, while achieving full-rank training (i.e., training with full-rank gradients of full-rank weights) to avoid inferior outcomes. First, we observe an interesting phenomenon during LLM training: the scaling impact of adaptive optimizers (e.g., Adam) on the gradient norm remains similar from low-rank to full-rank training. In light of this, we propose a *norm-based scaling* method, which utilizes the scaling impact of low-rank optimizers as substitutes for that of original full-rank optimizers to achieve this goal. Moreover, we find that low-rank optimizers may lead to potential loss spikes during training. To address this, we further put forward a *norm-growth limiter* to smooth the gradient. Extensive experiments on the pre-training and fine-tuning of LLMs show that Fira outperforms both LoRA and GaLore. Notably, for pre-training LLaMA 7B, our Fira uses $8\times$ smaller memory of optimizer states than Galore, yet outperforms it by a large margin.

## 1 Introduction

In recent years, Large Language Models (LLMs) have achieved remarkable advancements in various domains [2, 30, 10]. While the substantial increase in model size contributes significantly to these advancements, it also introduces considerable memory bottlenecks, especially for optimizer states [43]. For instance, pre-training a LLaMA-7B model from scratch [2] requires at least 58 GB memory, allocated as follows: 14GB for loading parameters, 14GB for weight gradients, 28GB for Adam [16] optimizer states, and 2GB for activations [43]. Notably, the optimizer states consume even more memory than the parameters themselves. To address this, low-rank training has demonstrated

---

*Corresponding author

[2]Training the model with a single batch size and maximum sequence length of 2048 under BF16 precision.

39th Conference on Neural Information Processing Systems (NeurIPS 2025).

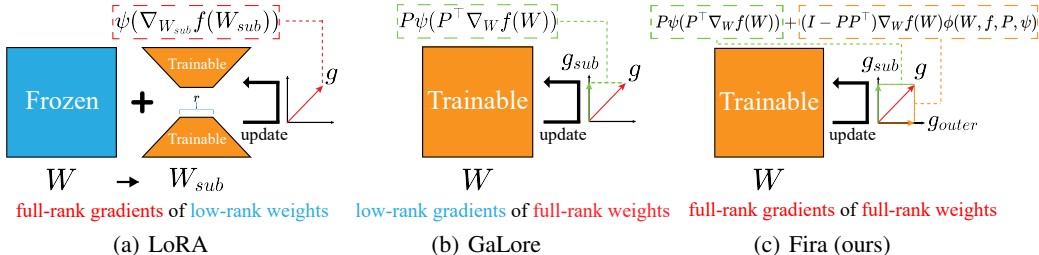

Figure 1: This analyses three types of memory-efficient approaches at a macro level.

its effectiveness in reducing the memory usage of the optimizer states by conducting training in a low-rank subspace [43, 13].

The current low-rank training methods can be broadly divided into two categories: weight matrix-based and gradient matrix-based low-rank decomposition. For the weight matrix decomposition methods, the most representative one is Low-Rank Adaptation (LoRA) [13], where its basic idea is to use low-rank matrices as decomposed representations of the pre-trained weights during training, as shown in Figure 1 (a). However, the optimization of LoRA is constrained in a low-rank subspace of the weights. This will inevitably cause the reduction of representation capacity, leading to sub-optimal outcomes [40, 35]. Although the variant ReLoRA [18] attempts to extend the application of LoRA from fine-tuning to pre-training, by periodically updating high-rank weights with multiple low-rank updates. It still requires full-rank weight training as a warm-up before low-rank training, thus rendering memory efficiency unachievable [43].

For the gradient matrix decomposition based methods, the typical one is the gradient low-rank projection (GaLore) proposed recently [43]. In contrast to LoRA, GaLore attempts to reduce the memory usage in optimizer states via decomposing the gradient matrix, as shown in Figure 1 (b). While GaLore supports the training of full-rank weights, it leverages only low-rank gradients, restricting them to a low-rank subspace. Consequently, any gradient information outside this subspace is lost, in contrast to training with full-rank gradients. Note that since these methods constrain LLM training to a low-rank subspace, this inevitably leads to sub-optimal results compared to full-rank training (i.e., training with full-rank gradients and full-rank weights). This raises the question: **Can we achieve full-rank training for LLMs while consistently maintaining a low-rank constraint?**

In light of this, we propose a new memory-efficient training framework for LLMs, called **Fira**, which, to the best of our knowledge, is the first to achieve **f**ull-rank tra**i**ning while consistently maintaining a low-**ra**nk constraint. To achieve this goal, a significant challenge is that the low-rank constraint makes it hard to preserve complete optimizer states (e.g., gradient momentum and variance) of full-rank weights in the commonly-used adaptive optimizer (e.g., Adam). As a result, the adaptive optimizer fails to correct the raw gradients without corresponding op-

Table 1: Cosine Similarity and MSE between low-rank and full-rank scaling factors at the matrix and column levels for pre-training LLaMA models ranging from 60M to 1B parameters, averaged over 10K steps. Detailed analyses are presented in Appendix D.

| Size | Matrix Level | | Column Level | |
|---|---|---|---|---|
| | Cosine Similarity | MSE | Cosine Similarity | MSE |
| 60M | 0.9922 | 3e-04 | 0.9273 | 3e-05 |
| 130M | 0.9901 | 2e-04 | 0.9046 | 2e-05 |
| 350M | 0.9893 | 1e-04 | 0.9174 | 1e-05 |
| 1B | 0.9795 | 2e-04 | 0.9229 | 1e-05 |

timizer states. Without this correction, adaptive optimization algorithms would degrade into simple stochastic gradient descent (SGD), leading to significantly reduced optimization performance [16, 39]. This point is further validated in Section 4.1 and Section 5.4.

Fortunately, we observe an interesting phenomenon during LLM training: at weight matrix level, the scaling factors [3] of low-rank gradients exhibit strong similarity to those of full-rank gradients. As illustrated in Table 1, Cosine Similarity and Mean Squared Error (MSE) between low-rank and full-rank scaling factors exhibit significant similarity (Cosine Similarity close to 1, while MSE close to 0). Detailed quantitative analyses of this similarity are presented in Appendix D.

---

[3]The scaling factor $\phi_t(R_t)$ is defined as $\frac{\|\psi(R_t)\|}{\|R_t\|}$, where $\|R_t\|$ is the norm of the raw gradient, $\|\psi(R_t)\|$ is the norm of the gradient corrected by the gradient correction function $\psi$ of the optimizer (e.g., Adam).

Based on this observation, we put forward a norm-based scaling method that utilizes the scaling factor of a weight matrix in low-rank training to replace the corresponding matrix's scaling factor in full-rank training. In this way, our scaling factor can also play a similar role in correcting the raw gradient, as adaptive optimizers do. Therefore, we can enable full-rank training while preserving the low-rank constraint.

However, we observe potential sudden increases in gradients caused by low-rank optimizers during the early stages of training, which can lead to spikes in training loss, as illustrated in Figure 3. This phenomenon primarily arises because our norm-based scaling method lacks the gradient stability compared to the original full-rank Adam optimizer. Such gradient instability can trigger substantial parameter updates, causing the loss function to reach a much higher value and undermining prior optimization efforts [11, 39]. Despite the use of gradient clipping techniques [25], this issue may not be adequately resolved, as shown in Figure 3. To this end, we propose a norm-growth limiter, which aims to smooth the gradient by restricting the magnitude of the gradient norm's increase. By employing our limiter, we adaptively convert sudden rises in gradients into gradual increases, thereby facilitating a smooth update that effectively mitigates the problem of loss spikes.

Our main contributions can be summarized as follows:

1. We propose Fira, a plug-and-play memory-efficient training framework of LLMs, constituting the first attempt to enable full-rank training consistently under the low-rank constraint. We will release the source code and package of our Fira into a Python library for easy use.

2. We design two components in Fira: a norm-based scaling strategy that leverages the scaling effects of low-rank optimizers to facilitate full-rank training, and a norm-growth limiter to address the issue of loss spikes by limiting the growth of gradient norm.

3. Extensive experiments on the pre-training and fine-tuning of LLMs across various parameter counts (60M, 130M, 350M, 1B, 7B) validate the effectiveness of Fira in both pre-training and fine-tuning tasks, surpassing both LoRA and GaLore.

## 2   Related Work

**Low-rank adaptation.** Low-Rank Adaptation (LoRA) has been introduced by [13] as an efficient fine-tuning method for LLMs. The core idea of LoRA is to freeze the pre-trained weights and introduce trainable low-rank matrices as decomposed representations of the pre-trained weights. In this way, the memory usage of training LLMs could be saved. Recently, a variety of methods by extending LoRA have been proposed to further improve the performance [41, 33, 35, 40, 9]. For instance, ReLoRA [18] is proposed to extend the application of LoRA from fine-tuning to pre-training. However, it still requires full-rank warm-up training before low-rank training, which prevents achieving memory efficiency. It is worth noting that while LoRA-based methods reduce memory usage by limiting training to a low-rank parameter subspace, they inevitably reduce representation capacity [35].

**Gradient projection.** Recent works [40, 35, 32] have indicated that LoRA may yield sub-optimal performance since its low-rank constraints in parameters. Inspired by traditional projected gradient descent methods [6, 4], GaLore [43] has been proposed recently to mitigate this problem. It enables full-parameter learning under low-rank constraints by projecting the gradient into a low-rank subspace, reducing memory usage for optimizer states. However, while GaLore allows memory-efficient full-parameter training, it confines the gradient to a low-rank subspace, discarding the portion outside the subspace and resulting in significant information loss.

**System-Based memory-efficient techniques.** Many system-based techniques have been developed to reduce memory usage in LLM training [5, 27]. However, most of these methods achieve memory efficiency by compromising either time or precision. Gradient checkpointing [5] is proposed to reduce memory usage by trading increased computational time for the re-computation of activations. Quantization [9] reduces memory consumption by using lower-bit data types but at the cost of model precision. Memory offloading [38, 27] reduces GPU memory usage by using non-GPU memory (e.g., CPU) as an extension. However, it introduces additional communication overhead, such as CPU-GPU transfer time. It's important to note that our proposed method is complementary to these approaches and can potentially be combined with them to further reduce memory usage.

# 3 Preliminaries

## 3.1 Regular Full-Rank Training

At time step $t$, we denote the full-rank weight matrix as $W_t \in \mathbb{R}^{m \times n}$. The full-rank gradient can be represented as $G_t = \nabla_W f_t(W_t) \in \mathbb{R}^{m \times n}$, where $f$ is the objective function. Then the regular full-rank training can be expressed as follows:

$$W_{t+1} = W_t - \eta \psi_t(G_t), \tag{1}$$

where $\eta$ is the learning rate, and $\psi_t$ is the gradient correction function of the optimizer (for vanilla SGD, $\psi_t(G_t) = G_t$). Instead of vanilla SGD, adaptive optimizers (e.g., Adam [16], AdamW [21]) are usually employed to correct the raw gradient for improving the training performance. However, this typically requires additional memory for storing optimizer states used in gradient correction. For instance, Adam [16] requires storing the optimizer states $M$ and $V$, which consume $2mn$ of memory. The gradient correction process is as follows:

$$M_t = \beta_1 M_{t-1} + (1 - \beta_1) G_t, \tag{2}$$

$$V_t = \beta_2 V_{t-1} + (1 - \beta_2) G_t^2, \tag{3}$$

$$\psi_t(G_t) = \frac{\sqrt{1 - \beta_2^t}}{1 - \beta_1^t} \cdot \frac{M_t}{\sqrt{V_t} + \epsilon}, \tag{4}$$

where all matrix operations are element-wise. $\beta_1$ and $\beta_2$ are Adam's hyper-parameters, and $\epsilon$ is a small constant (e.g., $1 \times 10^{-8}$) used for numerical stability. Since this regular full-rank training typically consumes a large amount of memory for training LLMs, many representative low-rank training methods, e.g., LoRA [13] and Galore [43], have been proposed to reduce memory usage in recent years.

## 3.2 Low-Rank Adaptation

The basic idea behind LoRA [13] is to use low-rank matrices as decomposed representations of the pre-trained weights during training, in order to reduce memory usage. Formally, LoRA freezes the full-rank weight matrix $W_0 \in \mathbb{R}^{m \times n}$ and incorporates two low-rank matrices $A_t$ and $B_t$ for training as:

$$W_t = W_0 + B_t A_t, \tag{5}$$

where $B_t \in \mathbb{R}^{m \times r}$, $A_t \in \mathbb{R}^{r \times n}$, and the rank $r < \min(m, n)$. While LoRA reduces memory usage by limiting training to a low-rank subspace of the weight, it inevitably diminishes the representation capacity of the weight matrix $W_t$ [35].

## 3.3 Gradient Low-Rank Projection

In contrast to LoRA, GaLore [43] utilizes a projection matrix $P_t \in \mathbb{R}^{m \times r}$ to project the full-rank gradient $G_t \in \mathbb{R}^{m \times n}$ to a low-rank gradient $R_t = P_t^\top G_t \in \mathbb{R}^{r \times n}$ ($m \leq n$)[4]. By doing so, the memory usage of optimizer states could be reduced. The parameter update in GaLore can be formulated as:

$$W_{t+1} = W_t - \eta P_t \psi_t(R_t), \tag{6}$$

where the projection matrix $P_t$ can be obtained through singular value decomposition (SVD) of $G_t$ and can be updated every $T$ step:

$$G_t = U \Sigma V^\top \approx \sum_{i=1}^{r} \sigma_i u_i v_i^\top, \quad P_t = [u_1, u_2, \ldots, u_r], \tag{7}$$

where $u_i$ is the $i$-th column vector of the left singular matrix $U$. By selecting the first $r$ columns of matrix $U$ that correspond to the largest singular values, the projection matrix $P_t$ effectively captures the most significant directions in the gradient space, leading to faster convergence [43]. The optimal switching frequency $T$ is usually set to be between $50$ to $1000$, and the additional computational overhead introduced by SVD is negligible ($< 10\%$), as stated in [43]. Since Galore restricts the gradient in the low-rank subspace, the gradient information outside this subspace is lost, leading to inferior performance.

---

[4]For simplicity, we assume $m \leq n$, following [43]. If $m > n$, $R_t = G_t Q_t \in \mathbb{R}^{m \times r}$, $Q_t \in \mathbb{R}^{n \times r}$.

## 4 Proposed Method

To achieve full-rank training under low-rank constraints, our framework, named Fira, consists of two important components: (i) a norm-based scaling method, enabling full-rank training by leveraging the scaling effects of adaptive optimizers; (ii) a norm-growth limiter, which restricts the growth of the gradient norm to prevent spikes in training loss. Next, we will elaborate on these two components.

### 4.1 Norm-Based Scaling

The low-rank constraint makes it challenging to record complete optimizer states for correcting raw gradients in full-rank training. Fortunately, we find an interesting phenomenon in LLM training: the scaling factor at the matrix level remains similar from low-rank training to full-rank training. Based on this observation, we propose a norm-based scaling strategy that approximately corrects the raw gradient, similar to adaptive optimizers, thereby enabling full-rank training.

**Challenge analysis.** Given the difficulty of incorporating trainable low-rank weights into LoRA to achieve full-rank weight training [43], we focus on investigating how to achieve full-rank gradient training by extending the gradient projection method, Galore, in this paper. In GaLore, the projection matrix $P_t \in \mathbb{R}^{m \times r}$ projects the full-rank gradient $G_t \in \mathbb{R}^{m \times n}$ of the full-rank weight $W_t \in \mathbb{R}^{m \times n}$, to the low-rank subspace gradient $R_t = P_t^\top G_t \in \mathbb{R}^{r \times n}$. Then, the gradient outside this subspace can be represented as: $(I - P_t P_t^\top)G_t = G_t - P_t R_t$. In other words, the full-rank gradient $G_t$ can be divided into two terms: $P_t R_t$ and $(G_t - P_t R_t)$.

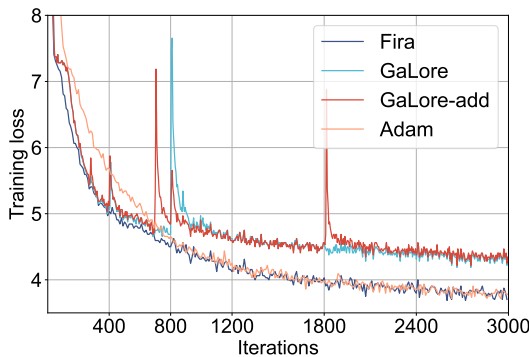

Figure 2: Training loss of different methods for pre-training LLaMA 60M on C4 dataset ($r/d_{model}$ = 16/256 and $T$ = 200).

In GaLore, the optimizer states only store the information of $R_t$ instead of $G_t$ to realize the low-rank constraint. The term of $(G_t - P_t R_t)$ is directly discarded in Galore due to the lack of corresponding optimizer states for correction in optimizers. This would lead to significant information loss especially when $r \ll d_{model}$, where $d_{model} = \min(m, n)$ is the full-rank dimension of models (This point can be verified in our experiment section, as illustrated in Figure 5. In Figure 5, the validation perplexity of GaLore significantly increases at $r = 4$ compared to $r = 128$ when $d_{model} = 256$, indicating a substantial loss of information and decreased training performance). Intuitively, to capture the information of $(G_t - P_t R_t)$, we can directly add it based on Eqn. 6 as follows:

$$W_{t+1} = W_t - \eta P_t \psi_t(R_t) - \eta(G_t - P_t R_t). \tag{8}$$

We denote the update strategy in Eqn. 8 as GaLore-add. However, as illustrated in Figure 2, GaLore-add exhibits almost no improvement compared to updates using Eqn. 6 in GaLore. This phenomenon primarily arises because the term of $(G_t - P_t R_t)$ doesn't have corresponding optimizer states for gradient correction. As a result, the optimization of $(G_t - P_t R_t)$ uses vanilla SGD, yielding suboptimal outputs. Besides, in GaLore-add, $P_t \psi_t(R_t)$ employs the Adam optimizer for training while $(G_t - P_t R_t)$ employs vanilla SGD. This gradient misalignment may also account for the lack of noticeable improvement.

**Similarity of scaling factor.** To tackle this challenge, we propose the concept of the *scaling factor*, which is defined as follows:

$$\phi_t(R_t) = \frac{\|\psi_t(R_t)\|}{\|R_t\|}, \tag{9}$$

where the scaling factor $\phi_t$ represents the magnitude of the correction applied by the adaptive optimizer to the gradient norm. Based on the scaling factor $\phi_t$, we observe an interesting phenomenon during LLM training: the scaling factors at the matrix level exhibit a high degree of similarity between low-rank and full-rank training (column level will be introduced later). As shown in Table 1, Cosine Similarity and MSE between low-rank and full-rank scaling factors exhibit significant similarity (Cosine Similarity close to 1, while MSE close to 0). Details analyses and additional experiments of this similarity are presented in Appendix D.

**Norm-based scaling.** Building on the above observation, we propose a norm-based scaling method that leverages the scaling factor of a weight matrix in low-rank training as a substitute for the corresponding factor in full-rank training:

$$W_{t+1} = W_t - \eta P_t \psi_t(R_t) - \eta \phi_t(R_t)(G_t - P_t R_t). \tag{10}$$

Through Eqn. 10, we can approximately correct $(G_t - P_t R_t)$ as adaptive optimizers do, so as to achieve full-rank training under low-rank constraints.

To further enhance our approach, we introduce a more fine-grained strategy for computing scaling factors in Eqn. 10 by considering each column of the weight matrix individually:

$$\phi_t(R_t)_i = \frac{\|\psi(R_{t,:,i})\|}{\|R_{t,:,i}\|}, \quad i = 1, 2, \ldots, n, \tag{11}$$

where $R_{t,:,i}$ is the $i$-th column of $R_t$, and $\phi_t(R_t)_i$ is the $i$-th scaling factor. As evidenced in Table 1, scaling factors computed at the column level also demonstrate strong similarity between low-rank and full-rank training, further validating the effectiveness of this fine-grained strategy. Additional theoretical analysis and experimental results on scaling factors are provided in Appendix E.

## 4.2 Norm-Growth Limiter

However, limited by low-rank constraints, we notice that there are still shortcomings in the norm-based scaling approach that need to be improved compared to the full-rank Adam. Specifically, potential sharp increases of the gradient may occur at the early stage of training, leading to spikes of training loss. As shown in Figure 3, Fira-w.o.-limiter (our method without using the proposed norm-growth limiter) experiences spikes in both gradient norm and training loss. In this section, we analyze the reasons for this issue and propose a norm-growth limiter that transforms abrupt gradient spikes into gradual, smooth increases, and then enhances the norm-based scaling method.

**Loss spike analysis.** There are two main reasons for the spikes: (i) Switching the projection matrix $P_t$ in gradient projection methods would cause instability during training. As illustrated in Figure 2, both GaLore and GaLore-add exhibit significant training loss spikes at integer multiples of $T = 200$ (i.e., the frequency of switching the projection matrix $P_t$). This instability occurs because, when switching projection matrices $P_t$, the optimizer retains states linked to the previous matrix, while the current input gradient uses a new projection matrix, leading to significant misalignment. Furthermore, as shown in Figure 2, GaLore-add also exhibits additional training spikes compared to Galore, reinforcing our earlier claim that directly incorporating $(G_t - P_t R_t)$ may introduce instability and hinder training; (ii) Maintaining the original direction of the raw gradient $(G_t - P_t R_t)$ may be insufficient for handling the sharp loss landscapes in LLM training, unlike Adam [39]. Due to space constraints, further analysis is provided in Appendix G.

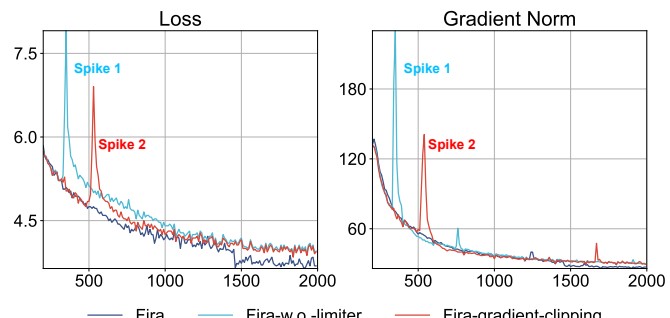

Figure 3: Training loss and gradient norm of three variants of Fira for pre-training LLaMA 60M.

**Addressing loss spikes.** To address this issue, a straightforward solution is to use gradient clipping techniques [25] to avoid loss spikes. However, clipping based on the absolute norm of gradient matrices fails to account for significant differences between them, leading to sub-optimal results. This point can also be verified in Figure 3 and Table 5. To this end, we propose a norm-growth limiter method that constrains the ratio of the current gradient norm to the previous step's norm to a fixed ratio $\gamma$ when the gradient norm increases:

$$\text{if } \frac{\|S_t\|}{\|S_{t-1}\|} > \gamma \text{ then } S_t \leftarrow \frac{S_t}{\|S_t\|} \cdot \gamma \|S_{t-1}\|, \tag{12}$$

where $\gamma$ is a threshold ensuring that the rate of gradient growth does not exceed this value. $S_t = \phi_t(R_t)(G_t - P_t R_t)$ is the corrected gradient by applying our norm-based scaling. This approach

limits the magnitude of gradient norm increases, converting sudden spikes into gradual rises and thus preventing loss spikes. Moreover, by constraining the relative increase of each gradient matrix's norm, our method is more flexible than the absolute norm clipping. As illustrated in Figure 2 and Figure 3, Fira with our proposed limiter improves the optimization performance without significant spikes.

**Discussion with other stabilization methods.** A variety of approaches have been developed to mitigate loss spikes and enhance training stability, including embedding normalization [17], gradient shrink on embedding layers [36], tensor-wise scaling [8]. While these methods contribute to stabilization, they either primarily concentrate on embedding stabilization (e.g., embedding normalization and gradient shrink), or employ static scaling (e.g., tensor-wise scaling). Compared to our norm-growth limiter, these techniques are insufficient in addressing the adaptive instability that arises from the substantial variations among different weight matrices, thereby inadequately resolving the issue of loss spikes in norm-based scaling. Further detailed analyses and experimental results regarding the norm-growth limiter are discussed in Section 5.4 and Appendix G.

### 4.3 Overall Algorithm

We present the overall algorithm of Fira with Adam in Algorithm 1 in Appendix A.1. Our main components, the norm-based scaling method, and the norm-growth limiter, are straightforward to implement, requiring only 3 additional lines of code. Moreover, Fira is a plug-and-play framework that can be easily integrated into the training process without requiring significant modifications. The plug-and-play Pytorch-like pseudo-code of Fira is provided in Appendix A.2.

Table 2: Comparison of different methods. Denote $W_t \in \mathbb{R}^{m \times n}$ ($m \leq n$), rank $r$.

| Method | Fira | GaLore | LoRA | Full-rank |
|---|---|---|---|---|
| Weights | $mn$ | $mn$ | $mn + mr + nr$ | $mn$ |
| Optimizer States | $mr + 2nr + 1$ | $mr + 2nr$ | $2mr + 2nr$ | $2mn$ |
| Gradients | $mn$ | $mn$ | $mr + nr$ | $mn$ |
| Full-Rank Gradients | ✓ | ✗ | ✓ | ✓ |
| Full-Rank Weights | ✓ | ✓ | ✗ | ✓ |
| Pre-Training | ✓ | ✓ | ✗ | ✓ |
| Fine-Tuning | ✓ | ✓ | ✓ | ✓ |

It's worth noting that compared to Galore, Fira only introduces one parameter $\|S_{t-1}\|$ for each weight matrix in the optimizer state, which is negligible, as shown in Table 2. Besides, in addition to the original hyper-parameters of optimizers and gradient projection methods, Fira only adds one hyper-parameter $\gamma$ in the norm-growth limiter, whose choice is not sensitive to the performance of Fira. The hyper-parameter $\gamma$ is set to 1.01 across all experiments, which consistently yields satisfactory results. Sensitivity analyses of hyper-parameter $\gamma$ to training performance are provided in Appendix H.

## 5 Experiments

In this section, we validate the effectiveness of Fira in pre-training and fine-tuning tasks of LLMs. In our experiments, we denote our method using the strategy of Eqn. 10 as Fira-matrix, and denote our method additionally using the column-wise strategy of Eqn. 11 as Fira.

### 5.1 Memory-Efficient Pre-training

**Experimental setup.** We follow the settings in Galore [43] to conduct the pre-training experiments. We compare Fira with GaLore [43], LoRA [13], ReLoRA [18], and full-rank training baselines. Adam optimizer is used to train all baselines and our method on the C4 dataset in the BF16 format. The settings of these baselines can be found in [43]. The dataset C4 is a colossal, cleaned version of Common Crawl's web crawl corpus, which is widely used in LLM pre-training [26]. Following [43], we utilize LLaMA-based architectures equipped with RMSNorm and SwiGLU activations [37, 29, 31]. As in [43], our training protocol excludes data repetition and spans a sufficiently large

Table 3: Comparison of different methods for pre-training LLaMA models of various sizes on the C4 dataset. We report validation perplexity (↓) with a memory estimate of total parameters, gradients and optimizer states. Results of all baselines are taken from [43]. $r$ refers to the rank and $d_{model}$ is the full-rank dimension of models.

|  | 60M | 130M | 350M | 1B |
|---|---|---|---|---|
| Full-Rank | 34.06 (0.48G) | 25.08 (1.01G) | 18.80 (2.74G) | 15.56 (10.40G) |
| **Fira** | **31.06** (0.36G) | **22.73** (0.77G) | **16.85** (1.90G) | **14.31** (6.98G) |
| GaLore | 34.88 (0.36G) | 25.36 (0.77G) | 18.95 (1.90G) | 15.64 (6.98G) |
| LoRA | 34.99 (0.44G) | 33.92 (0.99G) | 25.58 (2.12G) | 19.21 (7.36G) |
| ReLoRA | 37.04 (0.44G) | 29.37 (0.99G) | 29.08 (2.12G) | 18.33 (7.36G) |
| $r/d_{model}$ | 128 / 256 | 256 / 768 | 256 / 1024 | 512 / 2048 |
| Training Tokens | 1.1B | 2.2B | 6.4B | 13.1B |

dataset, encompassing a diverse array of model sizes (60M, 130M, 350M, 1B). To guarantee a fair comparison, we employ the same learning rate $0.01$ as used in GaLore and maintain the same rank $r$ for each model size. The detailed settings of pre-training are provided in Appendix B.1. We use 8 A100 80G GPUs to conduct pre-training experiments.

**Result analysis.** As shown in Table 3, Fira consistently outperforms low-rank training baselines by a large margin under the same rank constraint, and even surpasses full-rank training. Following [43], we estimate the memory reduction of the optimizer states via the same memory estimation method introduced in [43]. Detailed memory estimation methods are presented in C.1. Additional experiments on real memory usage and throughput are presented in C.2. From Table 3, our Fira saves 61.1% memory usage of the optimizer states when pre-training the LLaMA 1B architecture compared to full-rank training, while Fira achieves better results. Compared to full-rank training, Fira's superior performance may be attributed to the following reason: the gradient direction in the norm-based scaling method is determined by the current state, rather than by historical gradients in Adam. Therefore, Fira introduces a higher degree of randomness in training, which can enhance the model's ability to escape the local optima, leading to better training performance [44]. It's worth noting that previous work [42, 45] also reveals this phenomenon. Detailed analysis of this phenomenon is presented in Appendix F.

## 5.2 Scaling up to LLaMA 7B Pre-training

To validate the scalability of our method, we scale up by pre-training the LLaMA 7B model with the full-rank dimension $d_{\text{model}} = 4096$. We compare Fira with the GaLore baseline, which generally achieves the best performance among low-rank training base-lines, as shown in Table 3. As illustrated in Figure 4, our method demonstrates a significant improvement over GaLore for pre-training LLaMA 7B, while using an $8\times$ smaller rank (memory of optimizer states). This highlights Fira's effectiveness, suggesting it could be a viable solution for large-scale LLM pre-training.

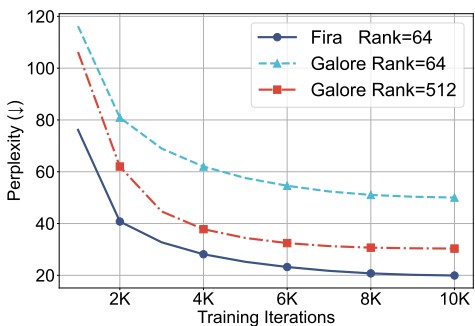

Figure 4: Pre-training LLaMA 7B with different methods on the C4 dataset.

## 5.3 Memory-Efficient Fine-Tuning

**Experimental setup.** Following [14], we perform the fine-tuning task to compare Fira with LoRA, GaLore, Flora [12], ReLoRA, Full-rank training, on the LLaMA-7B model for commonsense reasoning tasks. This task consists of eight sub-tasks, each with its own designated training and testing sets. Following the approach of [14], we combine the training datasets from all eight sub-tasks into a unified training set, while evaluating each sub-task individually using its respective testing dataset. In the fine-tuning task, the rank $r$ is set to 32 and the learning rate is set to 1e-4. The detailed

Table 4: Accuracy (↑) of various fine-tuning methods on eight commonsense reasoning datasets with LLaMA 7B. Results for all baseline methods, except GaLore, are taken from [14].

| Method | Memory | BoolQ | PIQA | SIQA | HellaSwag | WinoGrande | ARC-e | ARC-c | OBQA | Avg |
|---|---|---|---|---|---|---|---|---|---|---|
| **Fira** | 14.44G | 69.4 | **82.6** | **78.0** | 76.8 | **81.2** | **82.2** | 64.4 | **80.8** | **76.9** |
| GaLore | 14.44G | **69.5** | 82.0 | 75.1 | 32.2 | 18.0 | 80.7 | **65.8** | 78.0 | 62.7 |
| LoRA | 14.53G | 68.9 | 80.7 | 77.4 | **78.1** | 78.8 | 77.8 | 61.3 | 74.8 | 74.7 |
| ReLoRA | 14.53G | 68.9 | 81.2 | 77.8 | 46.0 | 79.4 | 80.2 | 64.2 | 79.6 | 72.2 |
| Flora | 14.44G | 50.1 | 77.5 | 74.2 | 53.8 | 45.5 | 79 | 64.6 | 74.8 | 64.9 |
| Full-rank | 56.00G | 64.2 | 68.1 | 68.0 | 42.3 | 66.5 | 55.6 | 43.9 | 60.0 | 58.6 |

settings of fine-tuning are provided in Appendix B.2. We adopt RTX 4090 GPUs for fine-tuning experiments.

**Result analysis.** As shown in Table 4, our Fira achieves the highest performance on 5 out of 8 datasets, demonstrating better or comparable performance compared to the baseline methods. Notably, GaLore struggles to adapt to the HellaSwag and WinoGrande datasets, resulting in a significant decline in scores. In contrast, our Fira adapts to these tasks well and achieves the highest scores on WinoGrande. In terms of memory efficiency, our method uses comparable or even less memory than the low-rank training methods LoRA and GaLore. These results illustrate the effectiveness of our method for the fine-tuning of LLMs.

## 5.4 Ablation Study

In this section, we conduct an ablation study to assess the effectiveness of each component in our method. We adopt the same settings in Section 5.1 for pre-training the LLaMA 60M model. We design six variants of our method for the ablation study: (1) **Fira-w.o.-scaling:** our Fira without using the scaling factor to correct the gradient (i.e., setting $\phi_t(R_t)$ to a fixed value of 1). (2) **Fira-matrix:** our Fira using the scaling factor at the matrix level instead of at the column level. (3) **Fira-w.o.-limiter:** our Fira without using norm-growth limiter to avoid training loss spikes. (4) **Fira-gradient-clipping:** our Fira using gradient clipping [25] to avoid loss spikes instead of our proposed norm-growth limiter. (5) **Fira-gradient-shrink:** our Fira using gradient shrink [36] instead of our proposed norm-growth limiter. (6) **Fira-tensor-wise-scaling:** our Fira using tensor-wise scaling [8] instead of our proposed norm-growth limiter.

Table 5: Ablation study on the C4 dataset.

| Method | Perplexity (↓) |
|---|---|
| Fira-w.o.-scaling | 37.06 |
| Fira-matrix | 31.52 |
| Fira-w.o.-limiter | 32.22 |
| Fira-gradient-clipping | 31.22 |
| Fira-gradient-shrink | 33.98 |
| Fira-tensor-wise-scaling | 33.81 |
| Fira | **31.06** |

Table 5 presents the results. It can be found that Fira outperforms Fira-w.o.-scaling, thereby demonstrating the effectiveness of our proposed norm-based scaling method for gradient correction. This also suggests that directly incorporating the raw gradient outside the subspace without correction will lead to sub-optimal results. Besides, Fira yields better performance than Fira-matrix, illustrating that a more fine-grained consideration of the scaling factor is beneficial. Furthermore, Fira demonstrates improved performance over Fira-w.o.-limiter, Fira-gradient-clipping, Fira-gradient-shrink, and Fira-tensor-wise-scaling, indicating the effectiveness of our proposed norm-growth limiter in addressing the issue of training loss spikes.

## 5.5 Performance under Varying Ranks

In this section, we illustrate the advantages of our Fira over Galore under a lower rank. We adjust various rank configurations within the set $\{4, 16, 64, 128\}$ and $d_{model} = 256$, and then assess the performance of pre-training the LLaMA 60M model on the C4 dataset as outlined in Section 5.1. The validation perplexity of Fira and GaLore after 10K steps across different ranks is depicted in Figure 5. From Figure 5, we can observe that Fira consistently surpasses GaLore across all rank configurations. Notably, even when the ranks are set very low (4 and 16), Fira still achieves performance comparable

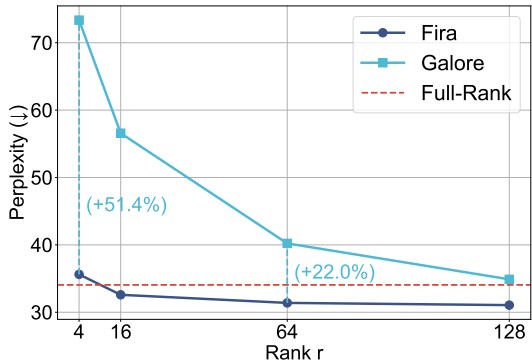

Figure 5: Validation perplexity of Fira and GaLore for varying ranks when pre-training LLaMA 60M on the C4 dataset with $d_{model} = 256$.

to full-rank training. In contrast, the performance of GaLore significantly declines in these cases. These results highlight the superiority of our proposed Fira at lower ranks and its effectiveness in reducing memory usage.

## 6 Conclusion

In this paper, we present a plug-and-play memory-efficient training framework for LLMs, called Fira, as the first attempt to facilitate full-rank training consistently under low-rank constraints. First, we find a notable phenomenon in LLM training: the scaling effect of adaptive optimizers on the gradient norm remains similar between low-rank and full-rank training. Building on this observation, we propose a norm-based scaling method that applies the scaling effect of low-rank optimizers in place of full-rank optimizers to facilitate full-rank training. This allows us to maintain the low-rank constraint within the optimizer while still benefiting from the advantages of full-rank training for improved performance. Additionally, we observe sudden spikes in gradient values during optimization, which can result in corresponding spikes in the training loss. To mitigate this, we propose a norm-growth limiter that smooths gradients by regulating the relative increase in gradient norms. Extensive experiments in both pre-training and fine-tuning of LLMs demonstrate the effectiveness of our proposed Fira.

## Acknowledgments and Disclosure of Funding

This work was supported by the NSFC under Grants U2441242.

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

# A  Fira Implementation

## A.1  Algorithm Pseudocode

---

**Algorithm 1** Fira with Adam

---

**Input:** Step size $\eta$, decay rates $\{\beta_1, \beta_2\}$, weight matrices $W \in \mathbb{R}^{m \times n}$ with $m \leq n$, rank $r$, switching frequency $T$, hyper-parameter of Galore $\alpha$, limiter threshold $\gamma = 1.01$.

$M_0, V_0 \in \mathbb{R}^{r \times n} \leftarrow 0, 0 \quad t \leftarrow 0$          {Initialize moving 1st, 2nd moment and step}

**repeat**

   $G_t \in \mathbb{R}^{m \times n} \leftarrow \nabla_W f_t(W_t)$         {Calculate full-rank gradients of full-rank weights}

   **if** $t \bmod T = 0$ **then**

      $U, \Sigma, V^\top \leftarrow \text{SVD}(G_t) \quad P_t \leftarrow U[:, : r]$      {Initialize the projection matrix every $T$ steps}

   **else**

      $P_t \leftarrow P_{t-1}$          {Reuse the previous projection matrix}

   **end if**

   $R_t, S_t \leftarrow P_t^\top G_t, (I - P_t P_t^\top) G_t$     {Divide gradients into two terms by gradient projection}

---

   $M_t \leftarrow \beta_1 M_{t-1} + (1 - \beta_1) R_t$        {$\psi_t(R_t)$: Apply Adam with low-rank gradients $R_t$}

   $V_t \leftarrow \beta_2 V_{t-1} + (1 - \beta_2) R_t^2$

   $N_t \leftarrow \frac{\sqrt{1 - \beta_2^t}}{1 - \beta_1^t} \cdot \frac{M_t}{\sqrt{V_t} + \epsilon}$

---

   $K \leftarrow [\frac{\|N_t[:, 1]\|}{\|R_t[:, 1]\| + \epsilon}, \frac{\|N_t[:, 2]\|}{\|R_t[:, 2]\| + \epsilon}, \cdots, \frac{\|N_t[:, n]\|}{\|R_t[:, n]\| + \epsilon}]$     {Norm-Based Scaling}

   $S_t \leftarrow [k_1 S_t[:, 1], k_2 S_t[:, 2], \cdots, k_n S_t[:, n]]$

   $S_t \leftarrow S_t \cdot \gamma / \max\{\frac{\|S_t\|}{\|S_{t-1}\| + \epsilon}, \gamma\}$       {Norm-Growth Limiter}

---

   $\tilde{G}_t \leftarrow \alpha \cdot (P_t N_t + S_t)$        {Project back and complete full-rank gradients}

   $W_t \leftarrow W_{t-1} - \eta \cdot \tilde{G}_t \quad t \leftarrow t + 1$       {Update the weight matrix}

**until** convergence criteria met

**return** $W_T$

---

## A.2  Plug-and-play Framework for Fira

---

**Algorithm 2** Plug-and-play framework for Fira, Pytorch-like.

---

1: **for** weight in model.parameters() **do**
2:    grad = weight.grad
3:    sub_grad, outer_grad = **project**(grad)         {Gradient projection.}
4:    sub_adapt = **adapt**(sub_grad)      {**Adaptive optimizer**, e.g., Adam, RMSProp.}
5:    outer_Fira = **Fira**(sub_grad, sub_adapt, outer_grad)     {Apply Fira to outer_grad.}
6:    weight_update = **project_back**(sub_grad) + outer_Fira      {full-rank training}
7:    weight.data += weight_update
8: **end for**

---

# B  Details of Experiments

## B.1  Detailed Pre-Training Setting

This section provides an overview of the LLaMA architectures and the hyper-parameters employed during pre-training. To ensure a fair comparison, we adopt the same settings as [43]. Table 6 presents the hyper-parameters of the LLaMA architectures across various sizes. For all architectures, we utilize a maximum sequence length of 256 and a batch size of 131K tokens. Furthermore, we implement a learning rate warm-up during the initial 10% of training steps and employ cosine annealing for the learning rate schedule, which decreases to 10% of the initial learning rate.

For all methods except Fira and GaLore, we tune the optimal learning rate from the set {0.01, 0.005, 0.001, 0.0005, 0.0001} across model sizes ranging from 60M to 1B, selecting their best validation

Table 6: Hyper-parameters of LLaMA architectures for pre-training.

| Params | Hidden | Intermediate | Heads | Layers | Steps | Data Amount (Tokens) |
|--------|--------|--------------|-------|--------|-------|----------------------|
| 60M    | 512    | 1376         | 8     | 8      | 10K   | 1.3 B                |
| 130M   | 768    | 2048         | 12    | 12     | 20K   | 2.6 B                |
| 350M   | 1024   | 2736         | 16    | 24     | 60K   | 7.8 B                |
| 1 B    | 2048   | 5461         | 24    | 32     | 100K  | 13.1 B               |
| 7 B    | 4096   | 11008        | 32    | 32     | 150K  | 19.7 B               |

perplexity to report. In contrast, both Fira and GaLore employ the same learning rate 0.01 and a subspace change frequency $T$ of 200 without tuning. Additionally, the scale factor $\alpha$ is considered a fractional learning rate [43]. Furthermore, a relatively large learning rate may result in spikes of training loss [43]. To address this issue, for models with a size of less than 1B, we set $\alpha$ to 0.25, while for models exceeding 1B, we adjust $\alpha$ to 0.0625.

## B.2 Detailed Fine-tuning Setting

We fine-tune the pre-trained LLaMA-7B model for commonsense reasoning tasks benchmark designed for LLM fine-tuning, which include eight sub-tasks [14]. Table 7 shows the hyper-parameter configurations.

Table 7: Hyper-parameter configurations of fine-tuning LLaMA-7B for Fira.

| Hyper-parameters | Setting |
|------------------|---------|
| Rank $r$         | 32      |
| $\alpha$         | 64      |
| Dropout          | 0.05    |
| Base optimizer   | Adam    |
| LR               | 1e-4    |
| LR Scheduler     | Linear  |
| Batch size       | 16      |
| warm-up Steps    | 100     |
| Epochs           | 3       |
| Where            | Q, K, V, Up, Down |

## C Details of Overhead Comparison

### C.1 Memory Estimates

Due to the difficulties associated with directly measuring GPU memory usage for a specific component, we estimate the memory requirements for weight parameters and optimizer states across various methods and model sizes, following GaLore [43]. This estimate is derived from the number of parameters and optimizer states in BF16 format. In particular, for the memory of parameters, we multiply the total number of parameters by 2; for the memory of optimizer states and gradients, we first calculate the total number of them according to Table 2 and then multiply this total number by 2.

### C.2 Additional Experiments on Real Overhead

We conduct additional comparisons regarding real memory usage and throughput of different memory-efficient training methods for both pre-training and fine-tuning. As illustrated in Tables 8 and 9, Fira achieves superior memory efficiency compared to full-rank training without significantly reducing throughput. Although Fira's throughput is slightly lower than that of other memory-efficient methods, it delivers exceptional performance. During pre-training, methods like LoRA necessitate maintaining additional higher-rank adapters (e.g. $r/d_{model} = 512/2048$) to achieve performance comparable

to full-rank training. However, in practice, maintaining these higher-rank adapters outweighs the benefits of fewer trainable parameters compared to full-rank training. This may be due to the fact that higher-rank adapters significantly increase computational demands during pre-training compared to fine-tuning (e.g. $r/d_{model} = 32/2048$), thus leading to more memory and less throughput. Furthermore, since full fine-tuning of LLaMA 7B's memory requirements exceeds the A100's 80GB capacity, we utilize DeepSeed's Zero2 technology to mitigate its memory usage.

Table 8: Real memory usage and normalized throughput when pre-training LLaMA 1B on the C4 dataset.

| Method | Fira | Galore | Flora | LoRA | ReLoRA | Full-rank |
|---|---|---|---|---|---|---|
| Memory (GB) | 54.6 | 54.6 | 54.5 | 59.0 | 59.0 | 58.5 |
| Normalized Throughput (%) | 94.2 | 95.9 | 95.9 | 67.4 | 67.4 | 100 |

Table 9: Real memory usage and normalized throughput when fine-tuning LLaMA 7B on common-sense reasoning datasets.

| Method | Fira | Galore | Flora | LoRA | ReLoRA | Full-rank |
|---|---|---|---|---|---|---|
| Memory (GB) | 23.4 | 23.4 | 23.3 | 23.7 | 23.7 | >80 |
| Normalized Throughput (%) | 156.1 | 201.1 | 210.3 | 232.8 | 232.8 | 100 |

# D Additional Quantitative Analysis of Scaling Factor Similarity

In this section, we analyze the similarities of scaling factors between low-rank and full-rank training at the matrix and column levels. Specifically, when analyzing similarity, we will form a vector of values for different scaling factors $(\phi_1, \phi_2, \ldots, \phi_k)$ ($\phi_i = \frac{\|\psi(R_t^{(i)})\|}{\|R_t^{(i)}\|}$). For the matrix level, we take the different weight matrices of the model as items (i.e., $R_t^{(i)} \in \mathbb{R}^{r \times n}$); for the column level, we take the columns in the weight matrix as items (i.e., $R_t^{(i)} \in \mathbb{R}^{r \times 1}$). We use the same low-rank setup as in Table 3. The models of different training runs are trained from the same random initialization. The only difference between them is the value of the rank hyperparameter (low-rank vs. full-rank).

## D.1 Spearman, Kendall, and Pearson Correlation Coefficients

In this section, we will employ Kendall's Tau correlation coefficient [1] and Spearman's rank correlation coefficient [28], Pearson's correlation coefficient [7] to evaluate the similarities of scaling factors between low-rank and full-rank training. We conduct our experiment by including all matrices of LLaMA models ranging from 60M to 1B. Then, we train these models and assess the similarity of scaling factors averaged over 10,000 steps. Additionally, to evaluate the effectiveness of a column-level fine-grained strategy for scaling factors, we perform a column-level quantitative similarity analysis. Due to the computational challenges posed by the large number of columns, we randomly sample 100 columns for each weight matrix for analysis. Specifically, in the LLaMA 1B model, over 10,000 columns are sampled.

Both Spearman, Kendall, and Pearson correlation coefficients range from -1 to +1. A coefficient of 1 signifies a perfect positive correlation, and -1 signifies a perfect negative correlation. The p-value helps us determine whether the observed correlation is statistically significant or if it could have occurred by random chance. For instance, a p-value less than 0.05 means there is less than a 5% probability that the observed correlation happened by chance if there was actually no correlation. Generally, a p-value below 0.05 suggests that a significant correlation exists. As shown in Table 10 , we can observe the significant similarity of scaling factors between low-rank and full-rank LLM training (all coefficients close to 1, while p-values close to 0). Thus, it is likely that the observed behavior is an inherent feature of LLM training, manifesting across a broad range of scenarios. This insight provides a robust experimental basis for our proposed norm-based scaling in Fira and helps explain its effectiveness.

Table 10: Spearman, Kendall, and Pearson correlation coefficients (p-values) at both the matrix and column levels for pre-training LLaMA models ranging from 60M to 1B parameters, averaged over 10,000 steps.

| Size | Matrix Level | | | Column Level | | |
|------|--------------|--------------|--------------|--------------|--------------|--------------|
| | Spearman | Kendall | Pearson | Spearman | Kendall | Pearson |
| 60M | 0.9972 (2e-62) | 0.9662 (7e-26) | 0.9891 (1e-46) | 0.9372 (0.0) | 0.7942 (0.0) | 0.8723 (0.0) |
| 130M | 0.9925 (2e-76) | 0.9409 (9e-37) | 0.9813 (2e-60) | 0.8698 (0.0) | 0.6830 (0.0) | 0.7805 (0.0) |
| 350M | 0.9770 (3e-113) | 0.8848 (5e-65) | 0.9766 (1e-112) | 0.9091 (0.0) | 0.7400 (0.0) | 0.8272 (0.0) |
| 1B | 0.9469 (1e-83) | 0.8249 (1e-56) | 0.9457 (6e-83) | 0.8331 (0.0) | 0.6513 (0.0) | 0.8112 (0.0) |

## D.2 Similarity Trends.

In this section, we conduct additional experiments on similarity trends of scaling factors through Cosine Similarity and Euclidean Distance using two rank settings: $r/d_{model} = 16/256$ and $r/d_{model} = 128/256$.

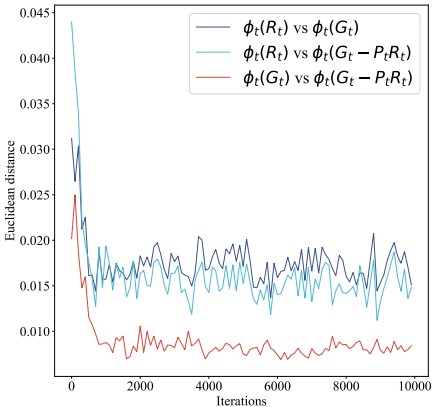

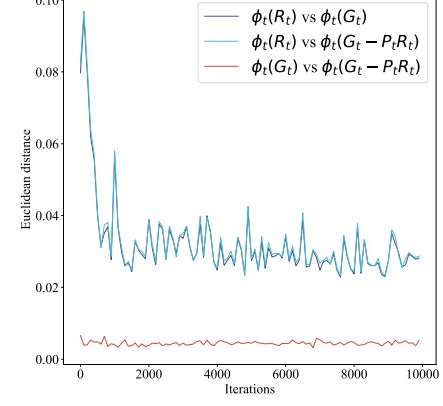

Figure 6: Euclidean Distance trends over training iterations ($r/d_{model} = 128/256$).

Figure 7: Euclidean Distance trends over training iterations ($r/d_{model} = 16/256$).

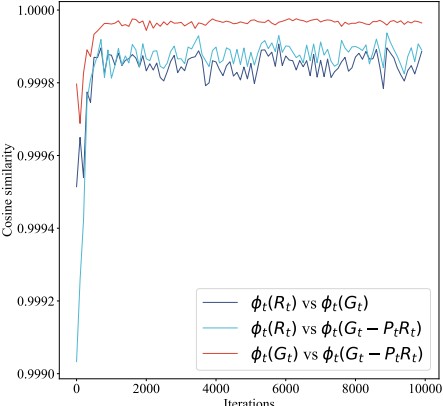

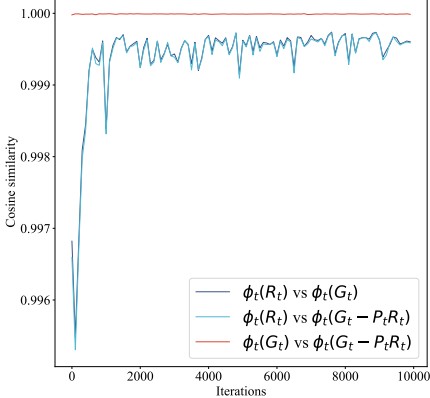

Figure 8: Cosine Similarity trends over training iterations ($r/d_{model} = 128/256$).

Figure 9: Cosine Similarity trends over training iterations ($r/d_{model} = 16/256$).

As shown in Figure 6, 7, 8, and 9, the similarity exhibits fluctuations during the initial training phase but achieves a relatively steady pattern with high similarity in the later iterations. Under lower rank setting $r/d_{model} = 16/256$, there is negligible reduction in similarity. Besides, $\phi_t(G_t)$ and $\phi_t(G_t - P_t R_t)$ demonstrate significantly higher similarity owing to their closely aligned dimensions.

## E  Theoretical Analysis

### E.1  Error Upper Bound for the Approximation of Scaling Factors

In Section 4.1, we use the scaling factors of low-rank gradients to approximate that of full-rank gradients. To quantify the effectiveness of this approximation, we will derive its error upper bound theoretically, and verify our analysis experimentally. The error of the approximation $\kappa(r)$ can be written as:

$$\kappa(r) = \left| \phi_t^2(G_t) - \phi_t^2(R_t) \right|, \tag{13}$$

where rank $r \leq n$, and $R_t = P_t^\top G_t \in \mathbb{R}^{r \times n}$. To simplify the proof, we consider that $r$ components $(g_1, \ldots, g_r)$ of low-rank gradients are directly sampled from $n$ components $(g_1, \ldots, g_n)$ of full-rank gradients. Under these conditions, the error can be rewritten as:

$$\kappa(r) = \left| \frac{\sum_{i=1}^n \psi_i^2(g_i)}{\sum_{i=1}^n g_i^2} - \frac{\sum_{i=1}^r \psi_i^2(g_i)}{\sum_{i=1}^r g_i^2} \right|. \tag{14}$$

**Assumption 1.** (Bounded Scaling Factors): we assume that the adaptive optimizer scales each gradient component $g_i$ by the scaling factor that lies within known bounds. Specifically, there exist constants $c_{\min}$ and $c_{\max}$ such that for all $i$:

$$c_{\min} \leq \left| \frac{\psi_i(g_i)}{g_i} \right| \leq c_{\max}. \tag{15}$$

This implies:

$$c_{\min}^2 \leq \frac{\psi_i^2(g_i)}{g_i^2} \leq c_{\max}^2. \tag{16}$$

**Theorem 1.** (Error Upper Bound for Approximation) Under the assumption that $\frac{\psi_i^2(g_i)}{g_i^2}$ are bounded between constants $c_{\min}^2$ and $c_{\max}^2$ for all components $i$, the approximation error $\kappa(r)$ satisfies:

$$\kappa(r) \leq \Delta_{\phi^2} \cdot \left( 1 - \frac{\|G_r\|}{\|G\|} \right), \tag{17}$$

where $\Delta_{\phi^2} \triangleq \sup_{g \neq 0} \frac{\psi^2(g)}{g^2} - \inf_{g \neq 0} \frac{\psi^2(g)}{g^2} = (c_{\max}^2 - c_{\min}^2)$ defines the squared scaling factor variation, $\|G\| = \sum_{i=1}^n g_i^2$, and $\|G_r\| = \sum_{i=1}^r g_i^2$.

*Proof.* We first define the following quantities: Total Gradient Norm $s_n = \sum_{i=1}^n g_i^2$, Partial Gradient Norm (first $r$ components) $s_r = \sum_{i=1}^r g_i^2$, Remaining Gradient Norm $s_{n-r} = s_n - s_r = \sum_{i=r+1}^n g_i^2$, Total Corrected Gradient Norm $S_n = \sum_{i=1}^n \psi_i^2(g_i)$, Partial Adjusted Gradient Norm (first $r$ components) $S_r = \sum_{i=1}^r \psi_i^2(g_i)$, Remaining Adjusted Gradient Norm $S_{n-r} = S_n - S_r = \sum_{i=r+1}^n \psi_i^2(g_i)$.

Then, our estimation error $\kappa(r)$ can be rewritten using these definitions:

$$\kappa(r) = \left| \frac{S_n}{s_n} - \frac{S_r}{s_r} \right|. \tag{18}$$

First, we rewrite the estimation error $\kappa(r)$ in terms of $S_r$, $S_{n-r}$, $s_r$, and $s_{n-r}$:

$$\kappa(r) = \left| \frac{S_r + S_{n-r}}{s_r + s_{n-r}} - \frac{S_r}{s_r} \right|. \tag{19}$$

Then, compute the difference in the numerator:

$$\kappa(r) = \left| \frac{(S_r + S_{n-r})s_r - S_r(s_r + s_{n-r})}{(s_r + s_{n-r})s_r} \right|. \tag{20}$$

Simplify the numerator, thus, the estimation error becomes:

$$\kappa(r) = \frac{|S_{n-r}s_r - S_r s_{n-r}|}{s_n s_r}. \tag{21}$$

After that, we factor out $s_{n-r}$:

$$\kappa(r) = \frac{s_{n-r}}{s_n} \cdot \left| \frac{S_{n-r}}{s_{n-r}} - \frac{S_r}{s_r} \right|. \tag{22}$$

From our bounded assumption, we have:

$$c_{\min}^2 \le \frac{S_r}{s_r} \le c_{\max}^2 \quad \text{and} \quad c_{\min}^2 \le \frac{S_{n-r}}{s_{n-r}} \le c_{\max}^2. \tag{23}$$

Therefore, the maximum possible difference between $\frac{S_{n-r}}{s_{n-r}}$ and $\frac{S_r}{s_r}$ is:

$$max\left( \left| \frac{S_{n-r}}{s_{n-r}} - \frac{S_r}{s_r} \right| \right) \le c_{\max}^2 - c_{\min}^2 = \Delta_{\phi^2}. \tag{24}$$

In addition, we have:

$$\frac{s_{n-r}}{s_n} = 1 - \frac{\sum_{i=1}^{r} g_i^2}{\sum_{i=1}^{n} g_i^2} = 1 - \frac{\|G_r\|^2}{\|G\|^2}. \tag{25}$$

Finally, the approximation error $\kappa(r)$ is bounded above by:

$$\kappa(r) \le \Delta_{\phi^2} \cdot \left( 1 - \frac{\|G_r\|^2}{\|G\|^2} \right). \tag{26}$$

$\square$

From this theory, we can find that the error upper bound on the approximation of scaling factors is mainly determined by two aspects, and we can verify them experimentally:

- Variability of Scaling Factor $\Delta_{\phi^2}$: This term represents the maximum variation in the scaling factors of different gradient components. For further validation, we designed *Fira-only-scaling*, a variant of Fira. It directly applies the low-rank scaling factors to the full-rank gradients by changing the Eqn. 10 from $W_{t+1} = W_t - \eta P_t \psi_t(R_t) - \eta \phi_t(R_t)(G_t - P_t R_t)$ to $W_{t+1} = W_t - \eta \phi_t(R_t) G_t$. In this way, we are able to exclude the influence of the original Adam term $P_t \psi_t(R_t)$ and better analyze the effectiveness of our approximation. As shown in Table 11, Fira-only-scaling (column-level) gains better performance than Fira-only-scaling (matrix-level) for its more fine-grained consideration of the scaling factor, which also means a smaller maximum variation $\Delta_{\phi^2}$.

- Effectiveness of Gradient Sampling $\left( 1 - \frac{\|G_r\|}{\|G\|} \right)$: This term represents the proportion of the gradients norm contributed by the sampled low-rank $r$ components from full-rank $n$ components. As shown in Table 12, we conducted ablation experiments *Fira-only-scaling-w.o.-svd*, i.e., Fira-only-scaling without SVD in low-rank gradient sampling. As we can see, SVD is capable of sampling more prominent low-rank gradients, which leads to a reduction in the upper bound of error and enhanced performance. Similarly, as shown in Table 13, employing a higher rank enables the sampling of a greater proportion of the gradients norm, resulting in reduced error upper bound and improved performance.

Table 11: Ablation on the level of scaling factors for the variant Fira-only-scaling.

| Level | Perplexity ($\downarrow$) |
|---|---|
| Column | 31.68 |
| Matrix | 32.05 |

Table 12: Ablation on SVD for the variant Fira-only-scaling.

| Method | Perplexity ($\downarrow$) |
|---|---|
| Fira-only-scaling | 31.68 |
| Fira-only-scaling-w.o.-svd | 32.22 |

Table 13: Ablation on rank for the variant Fira-only-scaling and full-rank Adam ($d_{model} = 256$).

| Rank | 4 | 16 | 64 | 128 | 256 (Full-rank) | Adam (Full-rank) |
|---|---|---|---|---|---|---|
| Perplexity ($\downarrow$) | 35.91 | 32.90 | 31.93 | 31.68 | 30.84 | 34.06 |

## E.2 Variance of Scaling Factors.

The variance of adaptive learning rates is significantly elevated during the early stage of training, often necessitating a warm-up to mitigate this variance and stabilize training [19]. As illustrated in Figure 10, the scaling factor in Fira exhibits a similar pattern, characterized by substantial variance during the early stage of training, which also necessitates a warm-up. However, the addition of an extra warm-up hyper-parameter for Fira would be inefficient. Therefore, it is crucial to investigate whether the original warm-up would have efficiently mitigated the variance in Fira. In the subsequent theoretical analysis, we show that, during the early training phase, the variance of the scaling factor of Fira is less than or equal to that of the adaptive learning rate. This finding suggests that the existing warm-up strategy is sufficient to mitigate the variance of Fira, thereby eliminating the need for an additional warm-up hyper-parameter.

Consider independent random vectors $\{\boldsymbol{g}^{(i)}\}_{i=1}^{n}$, where each $\boldsymbol{g}^{(i)} = (g_1^{(i)}, g_2^{(i)}, \ldots, g_t^{(i)})$. Here, the superscript $i$ indicates the index of the weight matrix to which the vector belongs, while the subscript $j$ (where $j$ ranges from 1 to $t$) denotes training iterations with each parameter. Following [19], we assume the adaptive learning rate of Adam $\psi(.) = \sqrt{\frac{1-\beta_2^t}{(1-\beta_2)\sum_{i=1}^{t}\beta_2^{t-i}g_i^2}}$, and $g_j^{(i)} \sim \mathcal{N}(0, \sigma^2)$ for all $i$ and $j$ in the early stage. Additionally, approximate the distribution of the exponential moving average as the distribution of the simple average, $p(\psi(.)) = p(\sqrt{\frac{1-\beta_2^t}{(1-\beta_2)\sum_{i=1}^{t}\beta_2^{t-i}g_i^2}}) \approx p(\sqrt{\frac{t}{\sum_{i=1}^{t}g_i^2}})$ [24], and then $\psi^2(.) \sim \text{Scale-inv-}\mathcal{X}^2(\rho, \frac{1}{\sigma^2})$.

**Theorem 2.** (Variance of Scaling Factors) In the early stages of training, if $\psi^2(\cdot) \sim \text{Scale-inv-}\mathcal{X}^2(\rho, \frac{1}{\sigma^2})$, and $g_j^{(i)} \sim \mathcal{N}(0, \sigma^2)^5$ for all $i, j$, then for all $\rho > 4$, the scaling factor $\phi^2 = \frac{\sum_{i=1}^{n}\psi_i^2(g_t^{(i)})^2}{\sum_{i=1}^{n}(g_t^{(i)})^2}$ satisfies $\text{Var}[\phi^2] \leq \text{Var}[\psi^2]$. If we approximate $\sqrt{\psi^2}$ and $\sqrt{\phi^2}$ to the first order, we have $\text{Var}[\phi] \leq \text{Var}[\psi]$.

*Proof.* We express $\phi^2$ as a weighted sum:

$$\phi^2 = \sum_{i=1}^{n} w_i \psi_i^2, \tag{27}$$

where the weights are defined as:

$$w_i = \frac{(g_t^{(i)})^2}{\sum_{j=1}^{n}(g_t^{(j)})^2}. \tag{28}$$

Each $w_i$ is a non-negative random variable satisfying $\sum_{i=1}^{n} w_i = 1$.

In the context of adaptive optimization algorithms like Adam, the squared gradients $\psi_i^2$ accumulate information from past iterations to adapt the learning rate for each parameter. With $\beta_2 = 0.999$,

---

[5]The assumption of a mean-zero normal distribution is valid at the outset of training, as the weights are sampled from normal distributions with a mean of zero [3]

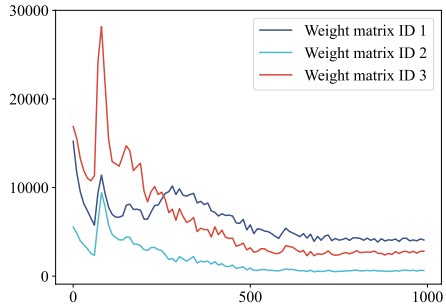

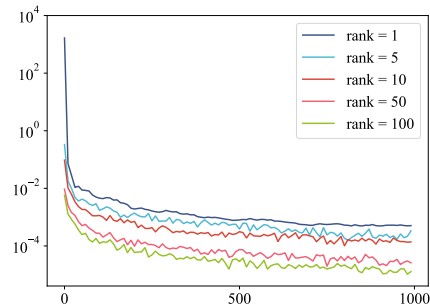

Figure 10: Scaling factor $\phi_t(R_t)$ during the early stage of training (1K iterations of total 10K iterations).

Figure 11: The simulation of variance of the scaling factor $\text{Var}[\phi]$ across different rank settings. The adaptive learning rate $\psi$ is equivalent to $\phi$ when the rank equals 1.

the moving average of the squared gradients places significant weight on historical data, making $\psi_i^2$ dependent mainly on past gradients, yielding:

$$\psi_i^2 \approx \psi_i^2(g_1^{(i)}, \ldots, g_{t-1}^{(i)}). \tag{29}$$

Since $\psi_i^2$ primarily depend on past gradients $g_1^{(i)}, \ldots, g_{t-1}^{(i)}$, and $w_i$ depend solely on the current gradients $g_t^{(i)}$, we can consider $\psi_i^2$ and $w_i$ to be independent random variables.

Consequently, we can express the variance of $\phi^2$ as:

$$\text{Var}[\phi^2] = \text{Var}\left[\sum_{i=1}^n w_i \psi_i^2\right]. \tag{30}$$

Using the law of total variance, we have:

$$\text{Var}[\phi^2] = \text{E}\left[\text{Var}\left[\sum_{i=1}^n w_i \psi_i^2 \mid w_1, \ldots, w_n\right]\right] + \text{Var}\left(\text{E}\left[\sum_{i=1}^n w_i \psi_i^2 \mid w_1, \ldots, w_n\right]\right). \tag{31}$$

Since $\psi_i^2$ are independent of the $w_i$, we find:

$$\text{E}\left[\sum_{i=1}^n w_i \psi_i^2 \mid w_1, \ldots, w_n\right] = \text{E}[\psi_i^2] \sum_{i=1}^n w_i = \text{E}[\psi_i^2], \tag{32}$$

$$\text{Var}\left[\sum_{i=1}^n w_i \psi_i^2 \mid w_1, \ldots, w_n\right] = \sum_{i=1}^n w_i^2 \text{Var}[\psi_i^2]. \tag{33}$$

Thus, the variance simplifies to:

$$\text{Var}[\phi^2] = \text{Var}[\psi^2] \text{E}\left[\sum_{i=1}^n w_i^2\right]. \tag{34}$$

The second term, $\text{Var}\left(\text{E}\left[\sum_{i=1}^n w_i \psi_i^2 \mid w_i\right]\right)$, is zero since $\text{E}[\phi^2 \mid w_i] = \text{E}[\psi_i^2]$ is constant.

Let $X_i = (g_t^{(i)})^2$, where each $X_i \sim \sigma^2 \chi_1^2$. Then, we can express the weights as:

$$w_i = \frac{X_i}{\sum_{j=1}^n X_j}. \tag{35}$$

Since $X_i/\sigma^2 \sim \chi_1^2$, and each $w_i$ is the ratio of $X_i$ to the sum of all $X_j$, the vector $(w_1, \ldots, w_n)$ follows a Dirichlet distribution with parameters $\alpha_i = \frac{\nu_i}{2} = \frac{1}{2}$, where $\nu_i = 1$ is the degrees of freedom of $\chi_1^2$.

For a Dirichlet distribution, the expected value of $w_i^2$ is given by:

$$\mathrm{E}[w_i^2] = \frac{\alpha_i(\alpha_i + 1)}{\left(\sum_{k=1}^n \alpha_k\right)\left(\sum_{k=1}^n \alpha_k + 1\right)}. \tag{36}$$

Substituting $\alpha_i = \frac{1}{2}$ and $\sum_{k=1}^n \alpha_k = \frac{n}{2}$ yields:

$$\mathrm{E}[w_i^2] = \frac{\frac{1}{2} \cdot \frac{3}{2}}{\frac{n}{2} \cdot \left(\frac{n}{2} + 1\right)} = \frac{3}{4} \cdot \frac{4}{n(n+2)} = \frac{3}{n(n+2)}. \tag{37}$$

Thus, summing over all $i$ gives:

$$\mathrm{E}\left[\sum_{i=1}^n w_i^2\right] = n \cdot \mathrm{E}[w_i^2] = \frac{3}{n+2}. \tag{38}$$

Finally, substituting this result back into the variance expression:

$$\mathrm{Var}[\phi^2] = \mathrm{Var}[\psi^2] \cdot \frac{3}{n+2}. \tag{39}$$

Since $n \geq 1$, it follows that:

$$\frac{3}{n+2} \leq 1, \tag{40}$$

which implies:

$$\mathrm{Var}[\phi^2] \leq \mathrm{Var}[\psi^2]. \tag{41}$$

Given $\rho > 4$ and $\psi^2(\cdot) \sim \text{Scale-inv-}\mathcal{X}^2(\rho, \frac{1}{\sigma^2})$, the variance of $\psi^2(\cdot)$ exists [19].

Since $\psi_i^2$ and $w_i$ are independent and $\sum_{i=1}^n \mathrm{E}[w_i] = 1$:

$$\mathrm{E}[\phi^2] = \mathrm{E}\left[\sum_{i=1}^n w_i \psi_i^2\right] = \sum_{i=1}^n \mathrm{E}[w_i]\,\mathrm{E}[\psi_i^2] = \mathrm{E}[\psi_i^2]\sum_{i=1}^n \mathrm{E}[w_i] = \mathrm{E}[\psi_i^2], \tag{42}$$

Thus, we have shown that:

$$\mathrm{Var}[\phi^2] \leq \mathrm{Var}[\psi^2], \quad \text{and} \quad \mathrm{E}[\phi^2] = \mathrm{E}[\psi_i^2]. \tag{43}$$

Follow [19], we approximate $\sqrt{\psi^2}$ and $\sqrt{\phi^2}$ to the first order [34]

$$\mathrm{Var}[\psi] \approx \frac{\mathrm{Var}[\psi^2]}{4\,\mathrm{E}[\psi^2]}, \quad \text{and} \quad \mathrm{Var}[\phi] \approx \frac{\mathrm{Var}[\phi^2]}{4\,\mathrm{E}[\phi^2]}. \tag{44}$$

which implies:

$$\mathrm{Var}[\phi] \leq \mathrm{Var}[\psi]. \tag{45}$$

$\square$

To further examine our theorem, we conduct simulations to calculate the variance of the scaling factor $\phi$ at ranks within the set $\{1, 5, 10, 50, 100\}$. The adaptive learning rate $\psi$ is equivalent to that of $\phi$ when the rank equals 1. As shown in Figure 11, the variance decreases as the rank increases, supporting our above theorem $\mathrm{Var}[\phi] \leq \mathrm{Var}[\psi]$. Furthermore, we observe a surprisingly large variance during the early stage, which corroborated our initial experiments. Consequently, we conclude that our method is efficient without requiring an additional warm-up.

## F  Potential Reasons for Fira's Enhanced Performance Compared to Full-Rank Optimizer

While our proposed method aims to approximate full-rank training, one might intuitively assume that the performance of full-rank Adam represents the upper bound. However, in previous experiments, we noticed that Fira may achieve performance that is comparable to or even better than full-rank training. This raises the question of why Fira outperforms full-rank Adam in certain scenarios. In this section, we first demonstrate that, despite its goal of approximating full-rank training, Fira is fundamentally

distinct from full-rank Adam. We then further substantiate this claim through experimental evidence. Finally, we explore potential reasons why Fira can achieve enhanced performance compared to full-rank Adam. Notably, the phenomenon where approximate optimizer correction outperforms the original full-rank optimizer has also been observed in prior work, such as [42, 45], consistent with our findings in Fira.

The approximation of low-rank to full-rank training in Fira is limited to scaling factors. Even in the case of full-rank training, Fira is not equivalent to the original Adam. To illustrate this, we introduce a variant of Fira called *Fira-only-scaling* in Appendix E.1. To isolate the influence of the original Adam term $P_t \psi_t(R_t)$, Fira-only-scaling directly applies low-rank scaling factors to the full-rank gradients by modifying Eqn. 10 from:

$$W_{t+1} = W_t - \eta P_t \psi_t(R_t) - \eta \phi_t(R_t)(G_t - P_t R_t), \qquad (46)$$

to:

$$W_{t+1} = W_t - \eta \phi_t(R_t) G_t. \qquad (47)$$

As shown in Table 13, higher-rank approximations of scaling factors lead to improved performance, validating the effectiveness of our approximation. But we can also find that full-rank Fira-only-scaling and Adam are not the same. Fira-only-scaling employs a weight-matrix-wise scaling factors $\phi_t(R_t)$ to adjust the raw gradients, whereas Adam uses element-wise adaptive learning rates.

As a result, in contrast to full-rank Adam, which is parameter-level adaptive, Fira applies adaptive strategy at matrix-level (column-level), while maintaining the original gradient direction within a matrix (column) (i.e., $\phi_t(R_t)G_t$). As a result, the gradient direction which is only determined by the current state can introduce a higher degree of randomness in training. Then, the randomness in training can enhance the model's ability to escape the local optima, thus leading to better generalization performance. This insight potentially explains why Fira could match or even surpass the full-rank Adam baseline.

To further argue this point, we conduct additional experiments on the comparisons of perplexity trends. We compare the perplexity trends of Fira, Fira-only-scaling, SGD, and Adam. As illustrated in 12 (a), the performance of vanilla SGD is significantly inferior, highlighting its inadequacy for directly training LLMs. As depicted in 12 (b) and (c), while Adam demonstrates faster convergence during the initial stages, both Fira and Fira-only-scaling achieve superior performance in the later stages. This may occur because Fira applies an adaptive strategy exclusively at the matrix level while preserving the original gradient direction within each weight matrix. As previously analyzed, Fira may introduce a higher degree of randomness in training and a better ability to escape the local optima, and then enable better generalization performance.

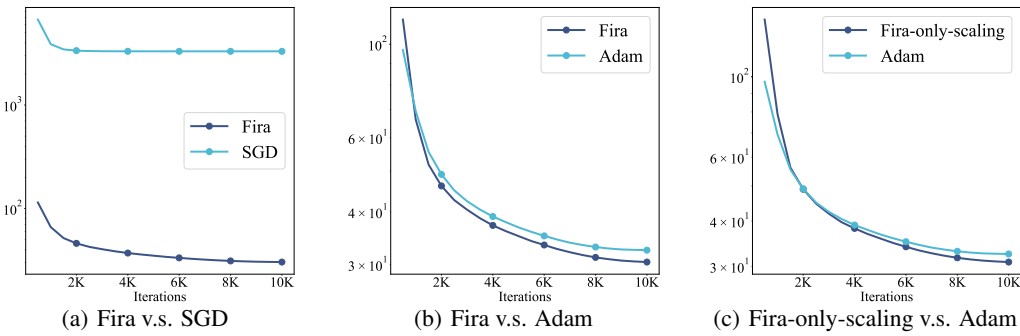

(a) Fira v.s. SGD      (b) Fira v.s. Adam      (c) Fira-only-scaling v.s. Adam

Figure 12: Comparisons of perplexity ($\downarrow$) trends for pre-training LLaMA 60M on C4 dataset.

## G    Additional Analysis of Spikes

Maintaining the direction of the raw gradient without correction might be unable to effectively deal with the steep loss landscapes of LLM training like Adam [39]. The steep loss landscapes are likely to cause abrupt increases in raw gradients. When the raw gradients increase abruptly, the gradients' norm after norm-based scaling may also increase abruptly, as illustrated in Figure 3. This arises from

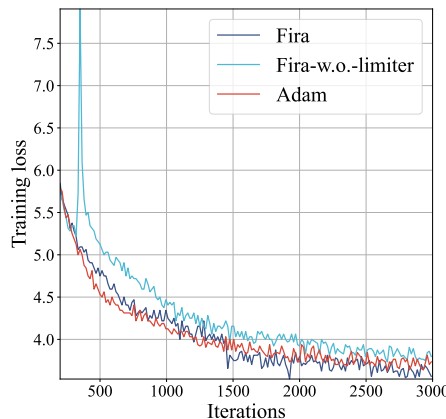
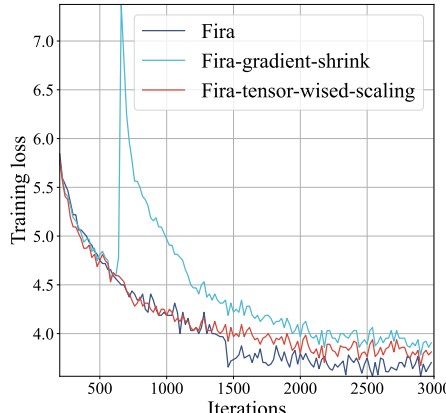

Figure 13: Training loss comparison of Adam, Fira, and Fira without limiter.

Figure 14: Training loss comparison of different gradient stabilization variants of Fira.

the fact that the norm-based scaling method only adjusts the average gradient norm of the gradient at the matrix level, failing to make fine-grained adjustments to each parameter, unlike the optimizer Adam. As a result, a significant parameter update may occur, undermining previous optimization efforts, i.e. training loss spikes [11, 39]. As shown in Figure 13, when we directly use Adam to pre-train the LLaMA model, there will be no loss spike. Our norm-growth limiter is mainly aimed at addressing the gradient stability capability that our norm-based scaling method lacks compared to Adam.

For more comprehensive comparisons of our norm-growth limiter, we design two additional gradient stabilization variants to solve the loss spike: Gradient Shrink [36] ($|S_t| = |S_t| \cdot \mu + |S_{t-1}| \cdot (1 - \mu)$), and Tensor-Wise Scaling [8] ($|S_t| = |S_t| \cdot \mu$), where $S_t = \phi_t(R_t)(G_t - P_t R_t)$ is the corrected gradient by applying our norm-based scaling. As shown in Figure 14 and Table 14, Fira outperforms other gradient stabilization methods. For further analysis, Gradient Shrink fails to solve the loss spike, while Tensor-Wise Scaling solves the loss spike but leads to sub-optimal results.

Table 14: Validation perplexity ($\downarrow$) of Fira across different gradient stabilization methods.

| Method | Ours | Gradient Shrink | Tensor-Wise Scaling | Gradient Clipping | Without Limiter |
|---|---|---|---|---|---|
| Perplexity ($\downarrow$) | **31.06** | 33.98 | 33.81 | 31.22 | 32.22 |

# H    Sensitivity Analyses of $\gamma$

Table 15: Validation perplexity ($\downarrow$) of Fira across different choices of $\gamma$.

| $\gamma$ | $\infty$ (w.o. limiter) | 1.1 | 1.01 | 1.001 | 1 |
|---|---|---|---|---|---|
| Perplexity ($\downarrow$) | 32.22 | 32.09 | 31.06 | 31.26 | 31.28 |

As shown in Table 15, the performance of Fira is not sensitive to the choice of $\gamma$. Provided that $\gamma \leq 1.01$, Fira can effectively mitigate spikes in loss, with only a marginal decrease in performance when $\gamma = 1$. As we can see, the setting of $\gamma = 1.01$, employed across all experiments in this paper, is highly effective. This value is neither too small to restrict the normal growth of the gradient nor too large to fail to limit sudden increases in gradient magnitude.

# I Discussion on Concurrent Works

Adam-mini [42] employs block-wise learning rates to replace Adam's element-wise learning rate based on the Hessian structure of neural networks. However, it only reduces memory usage for the second-order momentum while keeping the first-order momentum unchanged. In contrast, Fira reduces memory for both first-order and second-order momentum.

SlimAdam [15] uses layer-wise SNR analysis to replace second-moment tensors with their means for memory efficiency. However, like Adam-mini, it only focuses on the second-order momentum and is unable to reduce memory for both first-order and second-order momentum like Fira.

MicroAdam [23] compresses gradient information to save memory with theoretical convergence guarantees, but it is designed for fine-tuning workloads and cannot be adapted to LLM pre-training. However, Fira works for both LLM pre-training and fine-tuning.

AdaLomo [22] enhances low-memory optimization (LOMO) with adaptive learning rates to improve robustness and convergence. Unlike Fira, which focuses on enabling full-rank training under low-rank constraints, AdaLomo focuses on improving LOMO's convergence without involving the low-rank constraints.

FRUGAL [46] addresses gradient information loss outside the subspace in GaLore by directly adding these gradients outside the subspace via SGD or signSGD, without exploring more effective ways to utilize such gradients. In contrast, Fira explores the use of scaling factors to enable more effective updates for gradients outside the subspace. Additionally, FRUGAL provide theoretical convergence guarantees when using SGDM for subspace updates and SGD for updates outside the subspace.

GaRare [20] analyzes and compares the loss landscapes of low-rank and full-rank setting under gradient projection of GaLore, demonstrates that low-rank constraints improve the optimization landscape by avoiding sharp minima. This may help explain Fira's superior performance. Although GaLore's low-rank constraints enable a better optimization landscape, they discard significant gradient information outside the subspace, leading to suboptimal results. In contrast, Fira leverages low-rank constraints for a better optimization landscape while effectively capturing gradient information outside the subspace, resulting in superior performance.

# J Additional Experimental Comparison

To further demonstrate the generality of our method, we conduct additional experiments on a wider range of architectures (Gemma-7B, Mistral-7B), datasets (MMLU), and adaptive optimizer (Adagrad).

Table 16: Comparison results on MMLU tasks. Results of all baselines are taken from [45].

| Model | Methods | STEM | Social Sciences | Humanities | Other | Average |
|---|---|---|---|---|---|---|
| **Gemma-7B** | Full | 30.03 | 37.16 | 34.08 | 35.47 | 34.21 |
| | LoRA | 26.23 | 34.94 | 30.88 | 36.96 | 32.18 |
| | GaLore | 25.47 | 33.21 | 31.07 | 33.71 | 30.95 |
| | Fira | 29.03 | 35.27 | 32.40 | 36.52 | 33.26 |
| | APOLLO | 27.53 | 36.97 | 33.99 | 36.40 | 33.81 |
| **Mistral-7B** | Full | 52.40 | 72.95 | 55.16 | 69.05 | 61.67 |
| | LoRA | 52.13 | 72.46 | 55.05 | 68.77 | 61.41 |
| | GaLore | 51.87 | 72.82 | 54.94 | 69.49 | 61.56 |
| | Fira | 52.80 | 72.85 | 55.07 | 69.11 | 61.72 |
| | APOLLO | 51.63 | 73.12 | 54.90 | 69.58 | 61.58 |

Table 17: Validation perplexity ($\downarrow$) for pre-training LLaMA 60M on C4 dataset.

| Method | Fira+Adagrad | Adagrad | GaLore+Adagrad |
|---|---|---|---|
| Perplexity ($\downarrow$) | **43.11** | 103.85 | 88.87 |

As shown in Table 16 and 17, Fira remains competitive with all baselines across different model architectures, datasets, and optimizers, confirming its broad applicability.

To provide a comprehensive comparison against a broader range of baseline methods, we further extend our evaluation under the same experimental settings as in Tables 3 and 4.

Table 18: Pre-training LLaMA models on the C4 dataset. Validation perplexity ($\downarrow$) is reported.

|  | 60M | 130M | 350M | 1B |
|---|---|---|---|---|
| Fira | 31.06 | 22.73 | 17.03 | 14.31 |
| Adam-mini | 31.64 | 28.37 | 20.72 | 16.72 |
| SlimAdam | 31.22 | 24.70 | 18.12 | 15.42 |
| FRUGAL | 34.91 | 24.88 | 19.35 | 15.73 |

Table 19: Accuracy ($\uparrow$) on eight commonsense reasoning datasets when fine-tuning LLaMA 7B.

| Method | BoolQ | PIQA | SIQA | HellaSwag | WinoGrande | ARC-e | ARC-c | OBQA | Avg |
|---|---|---|---|---|---|---|---|---|---|
| Fira | 69.4 | 82.6 | 78.0 | 76.8 | 81.2 | 82.2 | 64.4 | 80.8 | 76.9 |
| Adam-mini | 69.8 | 78.3 | 65.1 | 38.6 | 80.3 | 78.2 | 64.5 | 80.4 | 69.4 |
| SlimAdam | 68.3 | 79.2 | 77.2 | 76.9 | 76.5 | 78.2 | 61.1 | 74.1 | 73.9 |
| FRUGAL | 68.2 | 78.5 | 76.1 | 77.0 | 77.5 | 80.1 | 63.5 | 76.8 | 74.7 |

As shown in Table 18, Fira demonstrates consistent superiority over these baselines when pre-training LLaMA models of different sizes. As shown in Table 19, when fine-tuning LLaMA 7B on commonsense reasoning datasets, Fira achieves the highest performance across most datasets, exhibiting superior average accuracy compared to these baselines.

## K   Robustness of Norm-based Scaling

As discussed in Adam-mini [42] and APOLLO [45], Adam's element-wise scaling contains significant redundancy. Using a shared scaling for a group of parameters, such as column-wise scaling in Fira-only-scaling, is sufficient to replace Adam's element-wise scaling. In this scenario, by aggregating historical gradient information across multiple parameters, column-wise norm-based scaling of Fira may be more robust than Adam's element-wise scaling when dealing with training data noise.

To empirically assess this robustness, we inject Gaussian noise into the gradients during pre-training and measure the resulting performance degradation.

Table 20: Validation perplexity ($\downarrow$) for pre-training LLaMA 60M on C4 dataset.

| Method | Without Noise | With Noise ($0.00001 \cdot \mathcal{N}(0, 1)$) |
|---|---|---|
| Fira-only-scaling | 31.68 | 34.64 |
| Adam | 34.06 | 445.47 |

As shown in Table 20, Fira-only-scaling exhibits slight performance degradation, while Adam's element-wise scaling shows significant performance degradation. This indicates that column-wise scaling is more robust.

## L   Limitations

This work presents Fira, a novel memory-efficient training framework that enables full-rank training of Large Language Models (LLMs) under low-rank constraints, significantly reducing memory usage while maintaining high performance. However, our current research primarily focuses on LLMs. In future work, we aim to extend the applicability of Fira to other domains, such as Multimodal Large Language Models (MLLMs) and diffusion models, to further broaden its impact across diverse machine learning tasks and architectures.

