# OpenReview forum: "Fira: Can We Achieve Full-rank Training of LLMs Under Low-rank Constraint?"
_NeurIPS.cc/2025/Conference — NeurIPS 2025 poster_

### Official Review · Reviewer_yQ8K · 2025-06-26

**Clarity:** 3
**Significance:** 3
**Originality:** 3
**Rating:** 5
**Confidence:** 4

**Summary:**

This paper introduces **Fira**, a training framework that enables **full-rank training of LLMs under low-rank memory constraints**. Unlike LoRA and GaLore that restrict training to low-rank subspaces and degrade performance, Fira uses a **norm-based scaling method** to approximate full-rank optimizer behavior using low-rank gradients. It also introduces a **norm-growth limiter** to stabilize training. Experiments show Fira consistently outperforms existing low-rank methods in both pre-training and fine-tuning, with significantly lower memory usage.

**Questions:**

1. While the empirical similarity between low-rank and full-rank scaling factors is clearly demonstrated, the theoretical justification (e.g., in Appendix E) remains limited and relies on simplifying assumptions. Could the authors provide a more rigorous theoretical explanation or clarify under what conditions such similarity is expected to hold?

2. The norm-based scaling update involves computing the residual term $G_t - P_t R_t$. It appears this term is used during the update step but not explicitly stored. Could the authors confirm whether this computation introduces any additional memory overhead compared to GaLore?

3. In Figure 5, Fira even outperforms full-rank training in some settings. This result appears consistent with findings in [1], which suggest that low-rank constraints can improve the optimization landscape by avoiding sharp minima. Could the authors discuss whether this phenomenon helps explain Fira’s superior performance? If so, it would be helpful to explicitly cite [1] and further analyze this connection.

[1] *On the Optimization Landscape of Low-Rank Adaptation Methods for Large Language Models*, ICLR 2025.

**Ethical Concerns:**

["NO or VERY MINOR ethics concerns only"]

**Final Justification:**

I believe this is a very good paper, though it doesn’t quite reach the level of having a groundbreaking impact. Therefore, I will maintain my decision to accept it.

**Limitations:**

The paper does not explicitly discuss the limitations of the proposed method.

1. The method has only been evaluated on LLaMA architectures, and its generalization to other popular LLM architectures (e.g., GPT, OPT, Mistral) remains unverified.
2. While the paper demonstrates strong results on LLaMA-7B, larger models have not been tested. It is unclear how Fira performs at even greater scales (e.g., 13B or 65B), where memory and stability challenges may differ.

**Paper Formatting Concerns:**

No formatting violations were detected.

**Quality:**

3

**Strengths And Weaknesses:**

Strengths:
1. The paper’s central insight—that the scaling factors of low-rank and full-rank gradients are highly similar—is both surprising and impactful. This observation is well-supported by quantitative evidence (cosine similarity, correlation coefficients), and it forms the basis for a practical and effective norm-based scaling method that approximates adaptive optimizer behavior without storing full-rank optimizer states.

2. A common but under-addressed issue in low-rank training is the occurrence of loss spikes, especially when switching projection subspaces. The proposed norm-growth limiter directly tackles this by adaptively smoothing gradient norm increases. Experiments demonstrate that this mechanism outperforms traditional gradient clipping and other heuristics, contributing to much more stable training.

3. The paper conducts extensive experiments across a wide range of LLaMA models (60M–7B) in both pre-training and fine-tuning settings. Fira consistently outperforms LoRA and GaLore in performance and memory usage, including an 8× reduction in optimizer memory when scaling to LLaMA-7B.

4. Fira is easy to integrate into existing training pipelines, requiring only minimal code changes.

Weaknesses:

1. The theoretical grounding for the norm-based scaling method relies on empirical similarity rather than formal guarantees. While the observed correlation is strong, a deeper theoretical explanation would strengthen the contribution.

2. The comparison scope omits other memory-efficient optimizers, such as Adafactor or 8-bit Adam. Including these baselines could better contextualize Fira’s benefits.

3. Only the LLaMA 7B model is used for large-scale validation. While the results are promising, testing on additional architectures (e.g., GPT, Mistral) would improve generality. That said, the limitation is understandable given the high computational cost.

---

> ### Author Rebuttal · Authors · 2025-07-30
>
> >Weakness1 & Question1: Theoretical explanation of similarity between low-rank and full-rank scaling factors.
>
> Thanks for your advice. As advised, we provide a more rigorous theoretical explanation of similarity between low-rank and full-rank scaling factors. In this theorem, we prove that the expected value of the ratio between full-rank and low-rank scaling factors is a constant, which demonstrates their similarity.
>
> Due to time constraints, we only present the final conclusion along with a brief proof. In the final version, we will further refine the proof details, complete the theoretical proof, and attempt to relax the conditions to derive more general conclusions.
>
> **Theorem.** (Similarity between low-rank and full-rank scaling factors). $G_t \in \mathbb{R}^{m \times n}$ is the full-rank gradient where $m\le n$, $P_t \in \mathbb{R}^{r \times m}$ is the projection matrix, $R_t=P_tG_t \in \mathbb{R}^{r \times n}$ is the projected low-rank gradient, where $r$ is the rank, and $\psi$ is the gradient correction function of the optimizer (e.g., Adam). The expected ratio of full-rank scaling factor $s_g=\frac{||\psi(G_t)||}{||G_t||}$ to low-rank scaling factors $s_r=\frac{||\psi(R_t)||}{||R_t||}$ is a constant $c$:
>
> $$\mathbb{E}[\frac{s_g}{s_r}]=\mathbb{E}[\frac{||\psi(G_t)||}{||G_t||} \cdot \frac{||R_t||}{||\psi(R_t)||}]=\mathbb{E}[\frac{||R_t||}{||G_t||}]\cdot\mathbb{E}[\frac{||\psi(G_t)||}{||\psi(R_t)||}] = c.$$
>
> *Proof:*
>
> As noted in GaRare [1]、FLoRA [2], the impact of projection matrix $P_t$ (sampled randomly from a normal distribution) on the final results is minimal. For the sake of simplifying the proof, we therefore assume $P_t \sim \mathcal{N}(0, \sigma^2)^{r \times m}$.
>
> For the first part $\frac{||R_t||}{||G_t||}$, it is equal to:
>
> $$\frac{||R_t||}{||G_t||}=\sqrt{\frac{||P_tG_t||^2}{||G_t||^2}}=\sqrt{\frac{||G_t^{\top}P_t^{\top}P_tG_t||^2}{||G_t||^2}}.$$
>
> For $(P_t^{\top}P_t)[i,j]$ where $i\neq j$，since all entries of $P_t$ are mutually independent:
>
> $$\mathbb{E}[(P_t^{\top}P_t)[i,j]]=0.$$
>
> For the case where $i=j$, $(P_t^{\top}P_t)[i,i]$ follows a chi-squared distribution, with an expected value of:
>
> $$\mathbb{E}[(P_t^{\top}P_t)[i,i]]=\sum_{k=1}^r P_{k,i}^2=r\sigma^2.$$
>
> From this, we derive:
> $$\mathbb{E}[P_t^{\top}P_t]=\sum_i \mathbb{E}[P_t^{\top}P_t[i,i]]=r\sigma^2\mathbb{I}_m.$$
>
> Thus:
> $$\mathbb{E}[\frac{||R_t||}{||G_t||}]=\mathbb{E}[\sqrt{\frac{||G_t^{\top}P_t^{\top}P_tG_t||^2}{||G_t||^2}}]=\sqrt{\frac{r^2\sigma^4||G_t||^2}{||G_t||^2}}=r\sigma^2.$$
>
> For the second part $\frac{||\psi(G_t)||}{||\psi(R_t)||}$, it is equal to:
> $$\frac{||\psi(G_t)||}{||\psi(R_t)||}=\frac{\sqrt{||\frac{M_t^G}{\sqrt{V_t^G}}||^2}}{\sqrt{||\frac{M_t^R}{\sqrt{V_t^R}}||^2}}=\frac{\sqrt{\sum^m_{i=1}\sum^n_{j=1}(\frac{M_t^G}{\sqrt{V_t^G}})^2[i,j]}}{\sqrt{\sum^r_{i=1}\sum^n_{j=1}(\frac{M_t^R}{\sqrt{V_t^R}})^2[i,j]}},$$
> where $M_t$ and $V_t$ are the first and second moments.
>
> Due to the element-wise division between $M_t$ and $V_t$, direct decomposition becomes challenging. To address this, we adopt a conclusion from Adam-mini [3]: the adaptive learning rates associated with $V_t$ in the Adam optimizer contain significant redundancy. Specifically, the $V_t$ values within a parameter block can be replaced by a single, well-chosen $V_t$.
>
> Based on this, we assume that different variances within a matrix can be replaced by the average of all $V_t$ values:
>
> $$\frac{||\psi(G_t)||}{||\psi(R_t)||}=\sqrt{\frac{\frac{\sum_{i=1}^{r} \sum_{j=1}^{n}  V^R_t[i,j]}{rn}}{\frac{\sum_{i=1}^{m} \sum_{j=1}^{n}  V^G_t[i,j]}{mn}}\cdot \frac{||M_t^G||^2}{||M_t^R||^2}}=\sqrt{\frac{m}{r}\cdot\frac{\sum_{i=1}^{r} \sum_{j=1}^{n}  V^R_t[i,j]}{\sum_{i=1}^{m} \sum_{j=1}^{n}  V^G_t[i,j]}\cdot \frac{||M_t^G||^2}{||M_t^R||^2}}.$$
>
> Assume that $beta_1$, $beta_2$ are exponential decay rates，for $\frac{||M_t^G||}{||M_t^R||}$，we have：
>
> $$\frac{||M_t^G||}{||M_t^R||}=||\frac{(1-\beta_1)\sum_{k=0}^{t-1}\beta_1^kG_{t-k}}{(1-\beta_1)\sum_{k=0}^{t-1}\beta_1^kR_{t-k}}||.$$
>
> For each term, result is similar to the derivation of the first part $\mathbb{E}[\frac{||R_t||}{||G_t||}]=r\sigma^2$. Thus:
>
> $$\mathbb{E}[\frac{||M_t^G||}{||M_t^R||}]=\frac{1}{r\sigma^2}.$$
>
> Moreover:
>
> $$\frac{\sum_{i=1}^{r} \sum_{j=1}^{n}  V^R_t[i,j]}{\sum_{i=1}^{m} \sum_{j=1}^{n}  V^G_t[i,j]}=\frac{\sum_{i=1}^{r} \sum_{j=1}^{n} (1-\beta_2)\sum_{k=0}^{t-1}\beta_2^kR_{t-k}[i,j]^2}{\sum_{i=1}^{m} \sum_{j=1}^{n} (1-\beta_2)\sum_{k=0}^{t-1}\beta_2^kG_{t-k}[i,j]^2}=\frac{(1-\beta_2)\sum_{k=0}^{t-1}\beta_2^k||R_{t-k}||^2}{(1-\beta_2)\sum_{k=0}^{t-1}\beta_2^k||G_{t-k}||^2}.$$
>
> By the same logic, for each term:
>
> $$\mathbb{E}[\frac{||R_{t-k}||}{||G_{t-k}||}]=r\sigma^2.$$
>
> From this, we derive:
> $$\mathbb{E}[\frac{\sum_{i=1}^{r} \sum_{j=1}^{n}  V^R_t[i,j]}{\sum_{i=1}^{m} \sum_{j=1}^{n}  V^G_t[i,j]}]=r^2\sigma^4.$$
>
> Thus:
>
> $$\mathbb{E}[\frac{||\psi(G_t)||}{||\psi(R_t)||}]=\sqrt{\frac{m}{r}\cdot r^2\sigma^4 \cdot \frac{1}{r^2\sigma^4}}=\sqrt{\frac{m}{r}}.$$
>
> Given that $\mathbb{E}[\frac{||R_t||}{||G_t||}]=r\sigma^2$，we have：
>
> $$\mathbb{E}[\frac{s_g}{s_r}]=\mathbb{E}[\frac{||\psi(G_t)||}{||G_t||} \cdot \frac{||R_t||}{||\psi(R_t)||}]=\mathbb{E}[\frac{||R_t||}{||G_t||}]\cdot\mathbb{E}[\frac{||\psi(G_t)||}{||\psi(R_t)||}]=r\sigma^2\sqrt{\frac{m}{r}}.$$
>
> It follows that $\mathbb{E}[\frac{s_g}{s_r}]$ is a constant.
>
> >Weakness2: Including other memory-efficient optimizers baselines.
>
> Thanks for your advice. As advised, we add comparisons with additional memory-efficient optimizers, including Adam-mini [3], SlimAdam [4].
>
> Table R1: Validation perplexity ($\downarrow$) for pre-training LLaMA 60M on C4 dataset.
> |Fira|Adam-mini|SlimAdam|
> |-|-|-|
> |31.06|31.64|31.22|
>
> As shown in Table R1, Fira outperforms these baselines.
>
> We will include more memory-efficient optimizers baselines in the final version.
>
> >Weakness3: Testing on additional architectures.
>
> Thanks for your advice. As advised, we conduct additional experiments on more architectures (Gemma-7B, Mistral-7B) to improve the generality.
>
> Table R2. Comparison results on MMLU tasks.
> |Model|Methods|STEM|SocialSciences|Humanities|Other|Average|
> |-|-|-|-|-|-|-|
> |Gemma-7B|Full|30.03|37.16|34.08|35.47|34.21|
> ||LoRA|26.23|34.94|30.88|36.96|32.18|
> ||GaLore|25.47|33.21|31.07|33.71|30.95|
> ||Fira|29.03|35.27|32.40|36.52|33.26|
> |Mistral-7B|Full|52.40|72.95|55.16|69.05|61.67|
> ||LoRA|52.13|72.46|55.05|68.77|61.41|
> ||GaLore|51.87|72.82|54.94|69.49|61.56|
> ||Fira|52.80|72.85|55.07|69.11|61.72|
>
> Results in Table R2 show that Fira remains competitive compared to baselines across different architectures.
>
> Due to time constraints, further experiments (e.g., pretraining experiments on more architectures) will be included in the final version.
>
> >Question2: Memory overhead of the residual term $G_t - P_tR_t$.
>
> Thanks for your comments. The residual term $G_t - P_tR_t$ is computed and used within the same update step and does not need to be stored for other steps. Thus, it introduces no additional memory overhead compared to GaLore.
>
> >Question3: Use findings in [1] to explain Fira’s superior performance.
>
> Thanks for your advice. We agree that the **findings in [1] help explain Fira’s superior performance** over full-rank training in some settings, as shown in Figure 5. Specifically, [1] analyzes and compares the loss landscapes of low-rank and full-rank setting，and then demonstrates that low-rank constraints improve the optimization landscape by avoiding sharp minima, which aligns with Fira’s results.
>
> Although GaLore's low-rank constraints enable a better optimization landscape, they discard significant gradient information outside the subspace, leading to suboptimal results. In contrast, **Fira leverages low-rank constraints for a better optimization landscape** while **effectively capturing gradient information outside the subspace**, resulting in superior performance.
>
> We will explicitly cite [1] and **further discuss with this work to clarify its impact on Fira's results in the final version**.
>
> >Limitations:
>
> 1. The method has only been evaluated on LLaMA architectures, and its generalization to other popular LLM architectures (e.g., GPT, OPT, Mistral) remains unverified.
>
> See response to Weakness3.
>
> 2. While the paper demonstrates strong results on LLaMA-7B, larger models have not been tested. It is unclear how Fira performs at even greater scales (e.g., 13B or 65B), where memory and stability challenges may differ.
>
> Thanks for your comments. Testing Fira's performance at even greater scales (e.g., 13B or 65B) requires significant computational resources. Due to time constraints, we will conduct experiments at greater scales to test our Fira in the final version.
>
> [1] On the Optimization Landscape of Low Rank Adaptation Methods for Large Language Models, ICLR'2025\
> [2] Flora: Low-rank adapters are secretly gradient compressors, ICML'2024\
> [3] Adam-mini: Use Fewer Learning Rates To Gain More, ICLR'2025\
> [4] When Can You Get Away with Low Memory Adam?

---

> > ### Comment · Reviewer_yQ8K · 2025-08-02
> >
> > Thank you for your comprehensive response. All of my concerns have been addressed. I believe this is a very good paper, though it doesn’t quite reach the level of having a groundbreaking impact. Therefore, I will maintain my decision to accept it.

---

### Official Review · Reviewer_K4iN · 2025-06-30

**Clarity:** 3
**Significance:** 3
**Originality:** 3
**Rating:** 5
**Confidence:** 4

**Summary:**

The authors present Fira, a method to handle low-rank optimizer scaling for full-rank training. The authors observe that the overall magnitude of Adam's update for a layer is very similar whether it's full-rank or low-rank. So Fira replaces Adam's per element update with an update that has the same direction as the full-rank gradient, but the magnitude is derived from the low-rank Adam stats.

Fira seems to have two components: a norm-based scaling strategy that leverages the scaling effects of low-rank optimizers to facilitate full-rank training, and a norm-growth limiter to address the issue of loss spikes by limiting the growth of gradient norm.

Fira introduces the concept of a scaling factor that represents the magnitude of the correction applied by the adaptive optimizer to the gradient norm. Since at the matrix level, it exhibits a high degree of similarity between low-rank and full-training training, Fira uses it as a norm-based scaling method that leverages the scaling factor of a weight matrix in low-rank.

Extensive experiments in both pre-training and fine-tuning of LLMs demonstrate the effectiveness of Fira.

**Questions:**

- Does it make sense to compare against system-level baselines such as ZeRO-offload in [23] or quantized optimizers?
- Is it fair to say that the method reduces the optimizer memory but activation and parameter memory can still dominate the total memory footprint?
- Downstream impact is somehow limited but it seems enough to demonstrate the applicability of the approach. What limitations do you forecast on vision or SFT or RLHF?
- How similar is conceptually and its effect the norm-growth limiter to gradient-clipping?

**Ethical Concerns:**

["NO or VERY MINOR ethics concerns only"]

**Final Justification:**

That's great. Thanks

**Limitations:**

yes

**Paper Formatting Concerns:**

No concerns

**Quality:**

3

**Strengths And Weaknesses:**

**Quality**

Strengths: The methodology looks good: the norm-based scaling rule and the limiter are both derived from concrete empirical observations about Adam's per-matrix norms, and the paper backs them up with ablations showing each part is indispensable (perplexity jumps from 31.06 to 37.06 without scaling, and to 32.22 without the limiter).

Experiments cover four model sizes up to 1 B parameters plus a 7 B scaling test, compare against strong baselines (LoRA, GaLore, ReLoRA, full-rank), and report both memory and accuracy metrics. The algorithm is easy to reproduce (three extra lines of PyTorch) and the authors release code.

Weaknesses: Real application evaluation is limited: pre-training uses only the C4 corpus and runs 10 K steps on the 7 B model, so long-horizon stability is untested.

**Clarity**

Strengths: The paper is easy to follow: Figure 1 contrasts LoRA/GaLore/Fira conceptually, Algorithm 1 gives step-by-step pseudocode, and spike analysis is illustrated with paired loss-and-norm plots. Notation is consistent and an appendix supplies proofs and implementation details.

**Significance**

Strengths: Optimizer-state memory is a real bottleneck for 7 B models and up. Reducing it by about 61% on a 1 B LLaMA while matching or beating full-rank perplexity is practically valuable. Because Fira is optimizer-agnostic, it looks very promising to be adopted widely.

Weaknesses: Memory footprint evaluation seems limited to the optimizer part.

**Originality**
Strengths: the idea that there’s some sort of norm scaling in Adam and that is preserved in low-rank and full-rank seems to be novel. This is proven empirically in Table 1 via cosine similarity.

Weaknesses: Fira is heavily based on Galore’s approach.

---

> ### Author Rebuttal · Authors · 2025-07-30
>
> Thanks for your valuable feedback! All suggested improvements will be developed and carefully incorporated into the final version.
>
> >Weakness1: Real application evaluation is limited: pre-training uses only the C4 corpus and runs 10 K steps on the 7 B model, so long-horizon stability is untested.
>
> + **Datasets for pre-training**.
>
> Thanks for your advice. Following the experiment setup in GaLore [1], we used the C4 dataset for pre-training. Due to time constraints, we will include pre-training experiments with additional datasets in the final version to further validate our approach.
>
> + **LLaMA 7B pre-training**.
>
> Thanks for your advice. To clarify, pre-training LLaMA 7B for 150K steps requires **significant computing resources**. For example, GaLore utilized  8-node training in parallel with a total of 64 A100 GPUs to complete the LLaMA 7B 150K-step pre-training experiments. If we use 8 A100s GPUs to complete them, it may take more than 2 months.
>
> At the same time, most of the concurrent work, e.g., Grass [2], FRUGAL [3], were unable to complete the pre-training of LLaMA 7B for 150K steps.
>
> As advised, we will conduct experiments with longer steps for pre-training LLaMA 7B in our future work to test the long-horizon stability.
>
> >Weakness2: Memory footprint evaluation seems limited to the optimizer part.
>
> Thanks for your comments. To clarify, in Tables 3 and 4 in our manuscript, **our memory footprint evaluation is not limited to the optimizer part**, but includes memory footprint evaluation of **total parameters, gradients, and optimizer states**.
>
> >Weakness3: Fira is heavily based on Galore’s approach.
>
> Thanks for your comments. While both Fira and GaLore target memory-efficient LLM training, they differ fundamentally: GaLore is restricted in low-rank gradient subspaces, thus completely failing to utilize the gradient information outside the subspace. However, Fira consistently employ full-rank gradients to train full-rank weights, achieving nearly unaffected performance despite reduced rank (PPL($\downarrow$) 31.06→32.59) compared to GaLore's severe performance degradation (PPL($\downarrow$) 34.88→56.57).
>
> Moreover, Fira is built upon a profound observation regarding the similarity of scaling factors between full-rank and low-rank training. It represents the first attempt of enabling consistent full-rank training under low-rank constraints. This breakthrough not only improves the memory-performance trade-off but also offers new insights for efficient LLM training.
>
> >Question1: Does it make sense to compare against system-level baselines such as ZeRO-offload in [23] or quantized optimizers?
>
> Thanks for your comments. To clarify, our method Fira and system-level baselines (like ZeRO-offload [23] or quantized optimizers) achieve memory efficiency memory efficiency from distinct perspectives. Our method, Fira, achieves memory efficiency **at the algorithm level**, which is **orthogonal to these system-level techniques**. In fact, Fira can be easily combined with them to further boost memory efficiency. For this reason, direct comparisons between them may not be entirely fair.
>
> >Question2: Is it fair to say that the method reduces the optimizer memory but activation and parameter memory can still dominate the total memory footprint?
>
> Thanks for your comments. To clarify, in LLM pre-training, especially when the number of parameters is very large, **optimizer memory is often more critical than activation and parameter memory**.
>
> For example, as mentioned in the Introduction in our manuscript, pretraining a LLaMA 7B model with FP16 using Adam requires 14GB for parameters, 24GB for optimizer states (2x the parameters memory), and only 2GB for activations. In this case, optimizer memory is significantly larger than activation and parameters memory. This effect becomes even more pronounced as model size increases.
>
> >Question3: Downstream impact is somehow limited but it seems enough to demonstrate the applicability of the approach. What limitations do you forecast on vision or SFT or RLHF?
>
> Thanks for your comments. As noted, we forecast the limitations when extending Fira to vision.
>
> Compared to LLMs, vision involves diverse modalities (e.g., images, videos, point clouds) and tasks (e.g., generation, detection, tracking). These differences may lead to distinct loss landscapes between LLMs and vision models. Thus, Fira may not work directly for vision. We would need to re-evaluate the phenomena and conclusions observed in LLM training, and adjust the method based on vision-specific characteristics.
>
> We will explore this point in our future work.
>
> >Question4: How similar is conceptually and its effect the norm-growth limiter to gradient-clipping?
>
> Thanks for your comments.
>
> Conceptually, the norm-growth limiter and gradient-clipping both address gradient instability but differ in focus. The norm-growth limiter adaptively controls the relative increase magnitude of the gradient norm, accounting for differences across different weight matrices. In contrast, gradient-clipping clip the gradients based on their absolute norm,  setting an upper limit to avoid overly large gradients.
>
> In terms of effect, as shown in our ablation study (Table 5 in our manuscript), the norm-growth limiter outperforms gradient-clipping.
>
> [1] GaLore: Memory-Efficient LLM Training by Gradient Low-Rank Projection, ICML'2024\
> [2] Grass: Compute Efficient Low-Memory LLM Training with Structured Sparse Gradients, EMNLP'2024\
> [3] FRUGAL: Memory-Efficient Optimization by Reducing State Overhead for Scalable Training, ICML'2025

---

### Official Review · Reviewer_FBSc · 2025-07-02

**Clarity:** 2
**Significance:** 3
**Originality:** 3
**Rating:** 5
**Confidence:** 4

**Summary:**

The paper proposes a new low-memory training technique for LLMs called Fira. This method reduces the memory footprint of Adam by storing optimizer states only for a low-rank approximation of the gradient, while still using the full-rank gradient at each step to improve performance. To achieve this, the authors not only use the low-rank gradient approximation with the standard Adam update (as done in low-rank methods like GaLore), but also utilize the remaining part of the full-rank gradient after applying column-wise or matrix-wise scaling to it based on the same low-rank optimizer states. The paper motivates this scaling approach and proposes a strategy to stabilize the optimization, specifically addressing the spikes in the loss curve that occur during training with this procedure. Empirical results demonstrate that Fira is effective compared to other low-memory methods in both pretraining and fine-tuning of LLMs, and it can even outperform full-rank Adam.

**Questions:**

All questions and concerns are detailed in the Strengths and Weaknesses section. The main weaknesses 1–3 are the most critical and will have the greatest impact on my final evaluation after the rebuttal.

**Ethical Concerns:**

["NO or VERY MINOR ethics concerns only"]

**Final Justification:**

**Initial submission.** The paper presented a new low-memory training method with strong performance in both pretraining and fine-tuning settings. However, it had issues with the clarity of the motivation for the method (W1), lacked analysis of the reasons for its effectiveness (W2), and provided only comparisons to basic baselines (W3).

**After discussion.** Most importantly, the authors provided a thorough comparison with stronger baselines (variations of memory-efficient Adam + FLoRa/GaRare) and additional results for other datasets, architectures, and optimizers. The motivation was also mostly clarified, and additional discussion and ablation experiments were provided to explain the method’s effectiveness (these should be further improved for the final version of the paper).

**Final recommendation.** The rebuttal and discussion addressed most of my concerns. While I think the motivation and the explanation of the method’s effectiveness could still be improved, the method itself is clearly effective even when compared to strong baselines. Therefore, I recommend the paper for acceptance and increase my score to 5.

**Limitations:**

I believe the limitations section should be more thorough and explicitly mention the limited understanding of the reasons behind the method’s benefits. Additionally, it should acknowledge that the experiments are limited to a single architecture, dataset, and adaptive optimizer.

**Paper Formatting Concerns:**

--

**Quality:**

3

**Strengths And Weaknesses:**

Strengths:
1. The paper addresses an important direction: low-memory optimization is crucial from a resource-efficiency perspective, yet the effectiveness of many existing methods remains limited in settings that require substantial training, such as pretraining or fine-tuning on highly dissimilar tasks.
2. The paper makes a reasonable effort to motivate all proposed modifications and supports them with a broad set of ablation experiments.
3. The proposed method demonstrates strong performance in both pretraining and fine-tuning settings. It not only outperforms other low-memory methods but also achieves better results than standard Adam in most of the cases.
4. Overall, while I have some concerns regarding the motivation and generalizability of the results, I find the empirical evidence convincing and believe the proposed method is effective.

Main weaknesses:
1. Motivation for scaling. Since scaling is a central component of the proposed method, I believe its motivation should be explained more clearly in the paper. Could you clarify how the comparison between scaling factors for the low-rank and full-rank gradients is performed? Are both computed within the same training run, and if so, what training procedure is used? My concern is that observing similar scaling factors along the trajectory of a single optimization procedure (either low-rank or full-rank) does not necessarily imply that the scaling factors are similar across the two procedures. As a result, it is not clear whether the proposed modification is expected to make low-rank training behave more like full-rank training.
2. Fira vs full rank. Continuing the previous point, it appears that the main accuracy gains of Fira do not result from a better approximation of full-rank training, but rather from generally changing the scaling procedure from element-wise to column-wise. This change is beneficial even in the full-rank setting, as indicated by the results for Fira-only-scaling in Table 13. However, the paper does not provide a clear explanation of why this scaling modification is effective. The discussion in the appendix about escaping local optima is vague and lacks depth. Moreover, column-wise scaling is not always better than element-wise scaling. Fira performs better than Fira-only-scaling, which suggests that element-wise scaling is helpful for the main components of the gradient but may be detrimental for others. Since the choice of scaling strategy is such an important factor in Fira’s performance, it should be analyzed in more detail in the paper.
3. Related works and baselines. From a practical perspective, my main concern is the limited discussion of other low-memory training techniques and the lack of comparison to them. For example, there is a line of research on low-memory Adam modifications, including Adam-mini (https://arxiv.org/abs/2406.16793), AdaLomo (https://arxiv.org/abs/2310.10195), MicroAdam (https://arxiv.org/abs/2405.15593), and SlimAdam (https://arxiv.org/abs/2503.01843 - this one which is a concurrent work but still relevant). These methods are clearly related to the proposed technique and therefore should be discussed in the related work section and included as baselines in the experiments. Additionally, GaLore-add, which is described as the first step of Fira’s modifications, appears very similar to FRUGAL (https://arxiv.org/abs/2411.07837), so this connection should be clearly stated. Overall, the baselines considered in the paper (LoRA, ReLoRA, GaLore) are quite basic. Including comparisons with current state-of-the-art low-memory training methods would be beneficial to better position the proposed method in context. I do not expect the proposed method to necessarily be state of the art, but such a comparison would provide valuable perspective.

Additional concerns, comments and questions:
1. Writing. The writing of the paper could be improved. In particular, the motivation for the proposed scaling technique is not clearly described in the main text. Even after reading the appendix, I still have some questions. Additionally, the comparison between matrix and column scaling is discussed somewhat inconsistently, which could be clarified.
2. Limited experimental results. All experiments in the paper are conducted using the same architecture (LLaMA) and include only one dataset for pretraining and one for fine-tuning. Additional experiments with different architectures and datasets would strengthen the generality of the claims. Moreover, based on the abstract and introduction, the paper appears to position the proposed modification as general for adaptive optimizers, while the experiments are performed only with Adam. The paper should either narrow its claims or provide results using other adaptive optimizers. Additionally, including full-rank experiments for LLaMA 7B pretraining would be beneficial to evaluate whether Fira consistently outperforms full-rank methods on larger-scale models.
3. Memory and compute. I am a bit confused about the memory usage comparisons in Tables 3 and 8. Could you please explain why Fira is expected to have a significant memory improvement according to Table 3, but shows almost no real improvement in Table 8? Also, why does Fira provide a substantial decrease in throughput compared to GaLore during fine-tuning but not during pretraining?
4. The scaling in Fira can be applied at the matrix or column level. Is the same true for the norm growth limiter, or is it always applied at the matrix level? If it is always applied at the matrix level, could you explain why?

---

> ### Author Rebuttal · Authors · 2025-07-30
>
> Thanks for your valuable feedback! All suggested improvements will be developed and carefully incorporated into the final version.
>
> >Weakness1: Details about motivation for scaling.
>
> Sorry for confusing you. Below, we provide clarified details on the comparison between scaling factors for the low-rank and full-rank gradients.
>
> 1. Are the scaling factors for the low-rank and full-rank gradients both computed within the same training run?
>
> **No, they are computed within different training runs**. As shown in Table 1 and Table 10 in our manuscript, we conducted multiple experiments under both low-rank and full-rank settings, computing and collecting the scaling factors independently for each experiment. Then, we averaged these scaling factors to assess their similarity between low-rank and full-rank settings.
>
> 2. How is the similarity of scaling factors between low-rank and full-rank settings calculated?
>
> For each training run, we compute column-wise scaling factors separately and combine them into a vector. After multiple experiments, based on two vectors computed under low-rank and full-rank settings, we can calculate their similarity using metrics like cosine similarity.
>
> >Weakness2: Fira vs full rank.
>
> Thanks for your comments. Below, we address the concerns raised from two perspectives. We will include a more detailed analysis in the final version.
>
> 1. **Why does column-wise Fira-only-scaling outperform element-wise Adam?**
>
> - **Robustness to data noise**
>
> As noted in Adam-mini [1] and APOLLO [2], Adam's element-wise scaling contains significant redundancy. Using a shared scaling for a group of parameters, such as column-wise scaling in Fira-only-scaling, is sufficient to replace Adam's element-wise scaling. In this scenario, by aggregating historical gradient information across multiple parameters, column-wise scaling may be more robust than Adam's element-wise scaling when dealing with training data noise.
>
> - **SGD-like updates improve generalization**
>
> Unlike Adam's element-wise scaling, column-wise Fira-only-scaling preserves original gradient direction within each column, leading to SGD-like updates. In many deep learning tasks, SGD yields better generalization [3, 4]. However, this benefit rarely extends to LLMs, as SGD struggles to train them [5]. As [5] explains, this is because SGD uses a uniform learning rate across all weight blocks, overlooking their inherent differences. However, Fira-only-scaling can take these differences into account by employing adaptive column-wise scaling. This may allow Fira to extend the strong generalization of SGD-like updates to LLMs. A similar perspective is presented in Apollo [2, Section 5.5].
>
> 2. **Why does Fira (column-wise + element-wise) outperform Fira-only-scaling?**
>
> - **Low-rank element-wise scaling benefits main gradient components**
>
> Prior works [6] find that low-rank element-wise scaling may improve the optimization landscape by avoiding sharp minima. In this case, low-rank element-wise scaling may be more effective than corresponding column-wise scaling, and can be helpful for the main components of the gradient. This may explain the improved performance of combination strategy Fira.
>
> - **More precise parameter grouping**
>
> While column-wise scaling reduces redundancy in element-wise Adam, it may not be optimal for all cases. For example, parameters may have diverse training dynamics within a certain column, requiring more fine-grained parameter grouping. Here, element-wise scaling may better handle the more fine-grained training dynamics thus complementing column-wise scaling.
>
> >Weakness3: Related works and baselines.
>
> Thanks for your advice. As advised, we will cite and discuss Adam-mini, AdaLomo, MicroAdam, SlimAdam, and FRUGAL in related works and add them as baselines in the experiments.
>
> **Related Work:**
>
> Adam-mini, which changes element-wise Adam to block-wise based on the Hessian structure of neural networks; AdaLomo, which enhances low-memory optimization (LOMO) with adaptive learning rates for better robustness to hyperparameters and convergence; MicroAdam, which compresses gradient information to reduce memory with theoretical convergence guarantees; and SlimAdam, which uses layer-wise SNR analysis to replace second-moment tensors with their means for memory efficiency.
>
> FRUGAL operates almost same as GaLore-add: both directly add the gradients discarded in GaLore, rather than exploring scaling factors for more effective updates of these gradients (as Fira does). But FRUGAL additionally provide theoretical convergence guarantees when using SGDM for subspace updates and SGD for updates outside the subspace.
>
> **Baselines:**
>
> We conduct additional experiments comparing Fira with Adam-mini, SlimAdam, and FRUGAL. As shown in Table R1, Fira achieves competitive performance against these low-memory training methods.
>
> Table R1: Validation perplexity ($\downarrow$) for pre-training LLaMA 60M on C4 dataset.
> |Fira|Adam-mini|SlimAdam|FRUGAL|
> |-|-|-|-|
> |**31.06**|31.64|31.22|34.91|
>
> Due to time constraints, we will include all mentioned baselines in the final version.
>
> >Question1: Writing.
>
> Sorry for confusing you.
>
> 1. Motivation for the proposed scaling technique
>
> As advised, we add more details in **response to Weakness1** to further explain the motivation for our proposed scaling technique.
>
> 2. Inconsistency of the comparison between matrix and column scaling
>
> In our manuscript, matrix scaling computes and applies scaling factors at matrix level, while column scaling operates at individual columns. In Table 5 of our ablation study, we compare Fira (column scaling) and Fira-matrix (matrix scaling), showing that the finer-grained column scaling performs better. Similarly, in Appendix E, Table 11, confirms that Fira-only-scaling (excluding the Adam term) performs better with column scaling than matrix scaling.
>
> >Question2: Limited experimental results.
>
> Thanks for your advice. As advised, we conduct additional experiments on more architectures (Gemma-7B, Mistral-7B), more datasets (MMLU), and more adaptive optimizers (Adagrad) to strengthen the generality of our claims. We will add more experiments in the final version.
>
> 1. Architectures and datasets
>
> Table R2. Comparison results on MMLU tasks.
> |Model|Methods|STEM|SocialSciences|Humanities|Other|Average|
> |-|-|-|-|-|-|-|
> |Gemma-7B|Full|30.03|37.16|34.08|35.47|34.21|
> ||LoRA|26.23|34.94|30.88|36.96|32.18|
> ||GaLore|25.47|33.21|31.07|33.71|30.95|
> ||Fira|29.03|35.27|32.40|36.52|33.26|
> |Mistral-7B|Full|52.40|72.95|55.16|69.05|61.67|
> ||LoRA|52.13|72.46|55.05|68.77|61.41|
> ||GaLore|51.87|72.82|54.94|69.49|61.56|
> ||Fira|52.80|72.85|55.07|69.11|61.72|
>
> As shown in Table R2, Fira remains competitive with the baselines across different architectures and datasets.
>
> 2. Adaptive optimizer
>
> Table R3: Validation perplexity ($\downarrow$) for pre-training LLaMA 60M on C4 dataset.
> |Fira+Adagrad|Adagrad|GaLore+Adagrad|
> |-|-|-|
> |**43.11**|103.85|88.87|
>
> As shown in Table R3, we conduct experiments on addtional adaptive optimizer Adagrad. Fira's strong performance demonstrates its compatibility with other adaptive optimizer.
>
> 3. Including full-rank experiments for LLaMA 7B pretraining.
>
> Due to high computational cost for pretraining LLaMA 7B, we will include it in the final version.
>
> >Question3: Queations about Fira's memory improvement and throughput decrease.
>
> Thanks for your comments.
>
> 1.Memory improvement
>
> Table 3's memory estimation only includes **parameters, gradients, and optimizer states**. However, Table 8's actual memory estimation includes additional parts, such as **activations, memory occupied by systems (e.g., temporary computation buffer)**, etc. These extra components will reduce the magnitude of the memory improvement in Table 8. Actually, the memory reduction is basically consistent between two tables: Table 3 shows a 3.42GB reduction (10.4GB - 6.98GB), and Table 8 shows a 3.9GB reduction (58.5GB - 54.6GB).
>
> 2.Throughput decrease
>
> Compared with GaLore, the decrease in Fira's throughput depends not only on time consumed by additional operations introduced by Fira, but also on **time consumed by the shared operations**. The longer time of shared operations, the smaller the decrease in Fira's throughput.
>
> Compared to fine-tuning, pre-training's shared operations dominate due to a larger batch size (16 for fine-tuning and 512 for pre-training). This explains why the throughput decrease of Fira is relatively small in pre-training.
>
> >Question4: Apply norm growth limiter at the matrix or column level.
>
> Thanks for your comments. **Like scaling in Fira, norm growth limiter can also be applied at the matrix or column level.** Below are the experimental results comparing matrix-level and column-level norm growth limiters.
>
> Table R4: Validation perplexity ($\downarrow$) for pre-training LLaMA 60M on C4 dataset.
> |Norm-growth limiter|Matrix-level|Column-level|
> |-|-|-|
> |PPL|31.06|31.09|
>
> As shown in Table R4, applying the norm growth limiter at the matrix or column level yields nearly identical results, indicating no significant difference between the two strategies.
>
> >Limitations.
>
> As advised, we further discuss the reasons behind Fira's benefits in **response to Weakness2**, and include additional experiments on additional architectures, datasets, and adaptive optimizer in **response to Question2**.
>
> We will further revise the Limitations section in the final version to provide a more comprehensive discussion of Fira's constraints.
>
> [1] Adam-mini: Use Fewer Learning Rates To Gain More, ICLR'2025\
> [2] APOLLO: SGD-like Memory, AdamW-level Performance, MLSys'2025\
> [3] Towards Theoretically Understanding Why SGD Generalizes Better Than ADAM in Deep Learning\
> [4] Improving Generalization Performance by Switching from Adam to SGD, NeurIPS'2020\
> [5] Why Transformers Need Adam: A Hessian Perspective, NeurIPS'2024\
> [6] On the Optimization Landscape of Low Rank Adaptation Methods for Large Language Models, ICLR'2025

---

> > ### Comment · Reviewer_FBSc · 2025-08-01
> >
> > Thanks for the detailed response!
> >
> > The main weaknesses I pointed out are not fully addressed – see comments/questions below. I’m especially concerned about the comparison to baselines in **Weakness 3**: as expected, these baselines are much stronger than those used in the paper, and the minimal comparison provided is not sufficient to support the claim that the proposed method is more effective. **I believe a proper comparison with these baselines is necessary before the paper can be accepted.**
> >
> > **Weakness 1: Details about the motivation for scaling.**
> >
> > Q1. Based on the rebuttal, the scaling factors for Table 1 are computed independently for each training run. I’m not sure I understand why the scaling factors would be correlated between two independent training runs from scratch. Suppose we have two training runs (regardless of whether they are low or full rank), why would we observe similar scaling factors for the same weight column if there is no alignment between the models? Since the models are trained from different random initializations, the weights/neurons should not be aligned.
> >
> > Q2. Could you elaborate on the connection between this empirical result and the theoretical justification in the reply to Reviewer yQ8K? As I understand it, the theoretical argument compares the low- and full-rank scaling factors for the same model checkpoint, while empirically, the scaling factors are computed on checkpoints from different training runs.
> >
> > **Weakness 2: FiRA vs full-rank.**
> >
> > While I appreciate the more thorough discussion on the effectiveness of the proposed scaling technique, I believe that since this is the main component of the proposed method, the paper should include not only discussion but also some experiments to validate this intuition. Moreover, the discussion presents arguments for column-wise scaling being both beneficial and detrimental, so it does not clearly explain why the specific combination used is effective in practice.
> >
> > Q3.  Prior works [6] find that low-rank element-wise scaling may improve the optimization landscape by avoiding sharp minima. – Could you please clarify where in [6] this claim is discussed? As far as I can tell, the paper does not explicitly address the type of scaling used.
> >
> > **Weakness 3: Related work and baselines.**
> >
> > As expected, the memory-efficient Adam variants are much stronger baselines than the basic LoRA/GaloRe used in the original submission. The performance of these new baselines is very close to that of the proposed method, so I don’t think a single experiment is sufficient to convincingly demonstrate the superiority of the proposed approach. Moreover, the proposed method and these baselines share similar ideas. I believe a more thorough experimental comparison and a clearer discussion of how the proposed method differs from these baselines are necessary before the paper can be accepted.
> >
> > **Additional concerns**
> >
> > I appreciate the explanations and additional experimental results provided in response to my additional questions. I strongly recommend that the authors incorporate them into the next version of the paper, especially the discussion on memory consumption and throughput.

---

> > > ### Author Response · Authors · 2025-08-05
> > >
> > > Thanks for your valuable feedback! Below, we will address your remaining concerns one by one.
> > >
> > > >Q1: Are the models trained from different random initializations?
> > >
> > > A1: Sorry for confusing you. To clarify, the models of different training runs are trained from the same random initialization. The only difference between them is the value of the rank hyperparameter (low-rank vs. full-rank). We will add this clarification to the final version.
> > >
> > > >Q2: Could you elaborate on the connection between this empirical result and the theoretical justification in the reply to Reviewer yQ8K?
> > >
> > > A2: Thanks for your comments. Theoretical derivation of the similarity in scaling factors across different training runs is challenging. Due to time constraints, our current theoretical justification is provided under simplified conditions, assuming that scaling factors are computed within the same training run.
> > >
> > > While this does not fully cover the setting of different training runs, it offers an initial intuition. In future work, we plan to relax these assumptions and extend the theoretical justification to account for different training runs, enabling a more general and rigorous explanation.
> > >
> > > >Q3：Prior works [6] find that low-rank element-wise scaling may improve the optimization landscape by avoiding sharp minima. – Could you please clarify where in [6] this claim is discussed?
> > >
> > > A3: Sorry for confusing you. To clarify, the "low-rank element-wise scaling" we mentioned refers to the optimization strategy Fira used in the low-rank subspace, which is consistent with the GaLore method.
> > >
> > > In [6], this idea is conveyed through the discussion that “GaLore is more likely to meet the condition of having no spurious local minima, whereas the full-rank method is less likely to avoid them,” and further notes that “the key to preserving GaLore’s favorable optimization landscape lies in projecting the gradient matrix into low-rank spaces.”
> > >
> > > Our wording was based on a summary phrasing from Reviewer yQ8K’s Question 3, which mentioned that “low-rank constraints can improve the optimization landscape by avoiding sharp minima.” We paraphrased this using "low-rank element-wise scaling," which we acknowledge could have caused confusion. We will add more details in the final version.

---

> > > ### Author Response · Authors · 2025-08-05
> > >
> > > >Q4: Include some experiments to validate our intuition.
> > >
> > > A4: Thanks for your advice. As advised, we add additional experiments to validate our intuition in "response to Weakness 2".
> > >
> > > 1. Robustness to noise.
> > >
> > > To evaluate the robustness of column-wise Fira-only-scaling and element-wise Adam scaling, we inject Gaussian noise into the gradients and measure the resulting performance degradation.
> > >
> > > Table R5: Validation perplexity ($\downarrow$) for pre-training LLaMA 60M on C4 dataset.
> > > |Method|Without Noise|With Noise ($0.00001\cdot\mathcal{N}(0, 1)$)|
> > > |-|-|-|
> > > |Fira-only-scaling|31.68|34.64|
> > > |Adam|34.06|445.47|
> > >
> > > As shown in Table R5, Fira-only-scaling exhibits slight performance degradation, while Adam’s element-wise scaling shows significant performance degradation. This indicates that column-wise scaling is more robust.
> > >
> > > 2. Column-wise Fira-only-scaling effectively solves the issue of SGD struggling to train LLMs.
> > >
> > > The difference between Fira-only-scaling and SGD is that Fira-only-scaling applies adaptive scaling to different columns based on their respective scaling factors, while SGD applies the same treatment to all columns.
> > >
> > > Table R6: Validation perplexity ($\downarrow$) for pre-training LLaMA 60M on C4 dataset.
> > > |Method|PPL|
> > > |-|-|
> > > |Fira-only-scaling|31.68|
> > > |SGD|$3\times 10^3$|
> > >
> > > As shown in Table R6, while SGD struggles to train LLMs, Fira-only-scaling achieves strong results. This indicates that Fira-only-scaling's column-wise scaling can effectively address the issue where SGD struggles to train LLMs.
> > >
> > > 3. Explain why the specific combination used is effective in practice.
> > >
> > > Fira uses GaLore's element-wise scaling in the subspace and column-wise scaling outside the subspace.
> > >
> > > This is because, as noted in [6], "Based on our theoretical analysis, the key to preserving GaLore’s favorable optimization landscape lies in projecting the gradient matrix into low-rank spaces." Unlike Adam, GaLore's element-wise scaling can preserve a favorable optimization landscape by projecting gradients into a low-rank subspace. As a result, Fira incorporates GaLore's element-wise scaling in the subspace to leverage such a beneficial optimization landscape.
> > >
> > > Table R7: Validation perplexity ($\downarrow$) for pre-training LLaMA 1B on C4 dataset.
> > > |Method|Fira（rank=512）|GaLore（rank=512）|Adam（rank=2048）|
> > > |-|-|-|-|
> > > |PPL|14.31|15.64|15.56|
> > >
> > > As shown in Table R7, despite significant gradient information loss outside the subspace, GaLore's low-rank element-wise scaling achieves performance comparable to that of full-rank Adam, confirming its effectiveness in the subspace.
> > >
> > > However, since gradients outside the subspace lack optimizer states, element-wise scaling cannot be directly applied. Therefore, outside the subspace, Fira employs column-wise scaling based on the corresponding scaling factors, effectively leveraging the substantial gradient information discarded by GaLore.
> > >
> > > In summary, Fira, on the one hand, retains the original low-rank element-wise scaling in the subspace to maintain a more favorable optimization landscape; on the other hand, it addresses significant gradient information loss in GaLore outside the subspace through column-wise scaling, ultimately achieving superior performance.
> > >
> > > Due to time constraints, we will include larger-scale and more comprehensive experiments, add more detailed discussions, and elaborate on more intuitions.

---

> > > ### Author Response · Authors · 2025-08-05
> > >
> > > >Q5: Comparison to baselines in Weakness 3.
> > >
> > > A5: Thanks for your advice. As advised, we add **a more thorough experimental comparison** and **a clearer discussion** of how the proposed method differs from these baselines.
> > >
> > > 1. **More thorough experimental comparison.**
> > >
> > > Following Table 3 and Table 4 in our manuscript's experiments, we **conduct additional experiments to strengthen the comparison with baselines**, including **pre-training LLaMA models of varying sizes** and **fine-tuning LLaMA 7B**.
> > >
> > > Table R8: Pre-training LLaMA models on the C4 dataset. Validation perplexity ($\downarrow$) is reported.
> > > |Model Size|60M|130M|350M|1B|
> > > |-|-|-|-|-|
> > > |Fira|31.06|22.73|17.03|14.31|
> > > |Adam-mini|31.64|28.37|20.72|16.72|
> > > |SlimAdam|31.22|24.70|18.12|15.42|
> > > |FRUGAL|34.91|24.88|19.35|15.73|
> > >
> > > As shown in Table R8, Fira demonstrates consistent superiority over these baselines when pre-training LLaMA models of different sizes.
> > >
> > > Table R9: Accuracy ($\uparrow$) on eight commonsense reasoning datasets when fine-tuning LLaMA 7B.
> > > |Method|BoolQ|PIQA|SIQA|HellaSwag|WinoGrande|ARC-e|ARC-c|OBQA|AVG|
> > > |-|-|-|-|-|-|-|-|-|-|
> > > |Fira|69.4|82.6|78.0|76.8|81.2|82.2|64.4|80.8|76.9|
> > > |Adam-mini|69.8|78.3|65.1|38.6|80.3|78.2|64.5|80.4|69.4|
> > > |SlimAdam|68.3|79.2|77.2|76.9|76.5|78.2|61.1|74.1|73.9|
> > > |FRUGAL|68.2|78.5|76.1|77.0|77.5|80.1|63.5|76.8|74.7|
> > >
> > > As shown in Table R9, when fine-tuning LLaMA 7B on commonsense reasoning datasets, Fira achieves the highest performance across most datasets, exhibiting superior average accuracy compared to these baselines.
> > >
> > > We will incorporate all the baselines you mentioned and conduct more comprehensive experiments in the final version.
> > >
> > > 2. **Clearer discussion.**
> > >
> > > + Adam-mini uses block-wise learning rates based on the Hessian structure of neural networks. However, it only reduces memory usage for the second-order momentum while keeping the first-order momentum unchanged. In contrast, Fira reduces memory for both first-order and second-order momentum.
> > >
> > > + SlimAdam uses layer-wise SNR analysis to replace second-moment tensors with their means for memory efficiency. However, like Adam-mini, it only focuses on the second-order momentum and is unable to reduce memory for both first-order and second-order momentum like Fira.
> > >
> > > + MicroAdam compresses gradient information to save memory with theoretical convergence guarantees, but it is designed for fine-tuning workloads and cannot be adapted to LLM pre-training. However, Fira works for both LLM pre-training and fine-tuning.
> > >
> > > + AdaLomo enhances low-memory optimization (LOMO) with adaptive learning rates to improve robustness and convergence. Unlike Fira, which focuses on enabling full-rank training under low-rank constraints, AdaLomo focuses on improving LOMO's convergence without involving the low-rank constraints.
> > >
> > > + FRUGAL addresses gradient information loss outside the subspace in GaLore by directly adding these gradients outside the subspace via SGD or signSGD, without exploring more effective ways to utilize such gradients. In contrast, Fira explores the use of scaling factors to enable more effective updates for gradients outside the subspace.
> > >
> > > Due to time constraints, we will refine this discussion to make it more comprehensive in the final version.
> > >
> > > Thank you for your appreciation of our responses to the additional questions. We will carefully incorporate all your suggested improvements, including those from the main weakness and additional questions, into the next version of the paper.
> > >
> > > [6] On the Optimization Landscape of Low Rank Adaptation Methods for Large Language Models, ICLR'2025

---

> > > > ### Comment · Reviewer_FBSc · 2025-08-08
> > > >
> > > > **Weakness 1.** Thanks for the clarifications. I would recommend that the authors clearly discuss the details of this experiment in the paper, since it provides the main motivation for the proposed method. Even after the discussion, I still have some technical questions about the experimental setup, because generally, even if two models are trained from the same initialization, they usually end up not aligned due to different sources of training noise (e.g., random batch order, data augmentations, and GPU non-determinism — see https://proceedings.mlr.press/v119/frankle20a.html).
> > > >
> > > > **Weakness 2.** I appreciate the additional discussion and experiments, and I think adding them to the paper would significantly improve the understanding of why the proposed method is effective.
> > > >
> > > > I find the results on noise robustness a convincing confirmation of why column-wise scaling can be beneficial. However, the discussion and results on why the specific combination of element-wise and column-wise scaling used on Fira is better than fully column-wise scaling are not fully convincing. I would recommend that the authors add a comparison of element-wise and column-wise scaling for the main components (i.e., GaLore vs. GaLore with column-wise scaling) to clearly demonstrate that element-wise scaling is important to achieving the benefits of Galore. Based on the current full-rank results (Adam vs. Fire-only-scaling), it is not clear if this is the case.
> > > >
> > > > **Weakness 3.** The new experimental results fully address my concern.
> > > >
> > > > **Final comment**
> > > >
> > > > The rebuttal and discussion addressed most of my concerns. While I think the motivation and the explanation of the method’s effectiveness could still be improved, the method itself is clearly effective even when compared to strong baselines. Therefore, I recommend the paper for acceptance and increase my score to 5.

---

### Official Review · Reviewer_f4Pe · 2025-07-05

**Clarity:** 3
**Significance:** 2
**Originality:** 3
**Rating:** 4
**Confidence:** 4

**Summary:**

This paper introduces Fira, a novel low-rank training framework that achieves the performance of full-rank training while maintaining the memory efficiency of low-rank methods. Fira restores full-rank training quality while keeping GaLore-style low-rank memory use. Starting from GaLore’s gradient-projection scheme, Fira restores GaLore’s discarded gradient component as a new term when updating, scales it with a norm-based factor that emulates adaptive optimizer corrections without extra state, and keeps training stable via a norm-growth limiter that caps each step’s gradient-norm increase. Experiment results show Fira outperforms full-rank, GaLore, LoRA and ReLoRA on LLaMA pre-training (60 M–1 B) and beats GaLore on LLaMA-7B using 8× smaller rank; fine-tuning on eight reasoning tasks also shows consistent gains.

**Questions:**

1. The paper emphasizes memory efficiency but provides limited quantitative analysis on computational overhead, especially the cost of additional operations introduced by Fira. Could the authors include a detailed breakdown of computational complexity or measured runtime overhead compared to baseline methods?

2. While comparisons with LoRA, GaLore, and ReLoRA are provided, newer methods involving tensor decompositions or other gradient compression techniques are not discussed. Could the authors elaborate on how Fira compares to other recent low-rank or gradient-efficient methods?

**Ethical Concerns:**

["NO or VERY MINOR ethics concerns only"]

**Limitations:**

yes

**Quality:**

3

**Strengths And Weaknesses:**

## Strengths
1. Novel norm-based scaling recovers discarded gradients in GaLore, which improve the performance without extra optimizer state storage.
2. Simple to implement but effective.
3. The experiments cover various model sizes (60M to 7B), demonstrate consistent performance gains in both pre-training and fine-tuning across multiple commonsense reasoning tasks, and include thorough and fair ablation studies. And also good result analysis, especially analysis in Appendix F.
4. Good error-bound analysis in Appendix E.
5. well-written

## Weaknesses
1. Although the experimental design is comprehensive, the comparison with more recent training methods is limited.

For instance:
Liu, Xu-Hui, et al. "On the Optimization Landscape of Low Rank Adaptation Methods for Large Language Models." The Thirteenth International Conference on Learning Representations. 2025.

How would this method work with the gradient random projection proposed above? Further discussion and an experiment are needed to draw a convincing conclusion.


2. No precise FLOPs since there is additional computing overhead.

---

> ### Author Rebuttal · Authors · 2025-07-30
>
> Thanks for your valuable feedback! All suggested improvements will be developed and carefully incorporated into the final version.
>
> >Weakness1: Additional comparison with GaRare [1].
>
> Thanks for your advice. As advised, we compare our method Fira with more recent training approaches, such as GaRare [1], and conduct experiments **integrating the gradient random projection proposed in GaRare with our Fira**.
>
> Table R1: Validation perplexity ($\downarrow$) for pre-training LLaMA 60M on C4 dataset.
> |Fira + gradient random projection|GaRare|GaLore|Fira|
> |-|-|-|-|
> |33.67|34.45|34.88|31.06|
>
> As shown in Table R1, Fira with gradient random projection proposed in GaRare outperforms GaRare thereby demonstrating the improvement in Fira remains effective under gradient random projection. Besides, Fira yields better performance than Fira + gradient random projection, indicating that the original SVD-based gradient projection may be more suitable than random projection under the Fira's framework.
>
> In the final version, we will revise the related work section by adding citations and a discussion of GaRare. Additionally, we will include more experiments and analyses on it to further strengthen the study.
>
> >Weakness2 and Question1: Additional quantitative analysis on computational overhead.
>
> + Precise FLOPs
>
> Thanks for your advice. As advised, we additionally measure the precise FLOPs of GaLore and Fira.
>
> Table R2: Precise FLOPs of GaLore and Fira.
> ||Fira|Galore|
> |-|-|-|
> |FLOPs|25.78G|21.47G|
>
> + Measured runtime overhead
>
> Thanks for your advice. As advised, we additionally measure the  real computational time of optmizaion step.
>
> Table R3: Optimizer step time (in seconds) across LLaMA-1B with a sequence length of 1024 on a single A100 GPU.
> |Model Size|GaLore|Fira|
> |-|-|-|
> |1B|1.440s|1.490s|
>
> As shown in Table R2 and Table R3, compared to GaLore, Fira only slightly increases the computational overhead, which is acceptable.
>
> >Question2: Comparisons with other recent low-rank or gradient-efficient methods.
>
> Thanks for your advice. As advised, we additionally elaborate on how Fira compares to other recent low-rank or gradient-efficient methods, e.g., GaRare [1] and FLoRA [2].
>
> Table R4: Pretraining LLaMA models on the C4 dataset. Validation perplexity ($\downarrow$) is reported.
> |Model Size|60M|130M|350M|1B|
> |-|-|-|-|-|
> |Fira|31.06|22.73|17.03|14.31|
> |GaRare|34.45|25.34|19.35|15.88|
> |FLoRA|36.97|30.22|22.67|20.22|
>
> As shown in Table R4, Fira achieves better performance compared to these methods across different model sizes.
>
> [1] On the Optimization Landscape of Low Rank Adaptation Methods for Large Language Models, ICLR'2025\
> [2] Flora: Low-rank adapters are secretly gradient compressors, ICML'2024

---

### Decision · Program_Chairs · 2025-09-17

**Decision:**

Accept (poster)

**Comment:**

The authors tackle the important problem of reducing the memory usage required for training or fine-tuning large language models (LLMs). They propose Fira, a plug-and-play framework for LLM training that maintains low-rank memory efficiency while enabling full-rank training through norm-based scaling. The core idea is to leverage the scaling effects of low-rank optimizers as substitutes for traditional full-rank optimizers.

Experimental results demonstrate that Fira outperforms both LoRA and GaLore in pre-training and fine-tuning, achieving superior performance with lower memory costs. Experiments include challenging reasoning tasks and a comprehensive ablation study. The paper is well-written, presents a novel idea, and offers improvements in a highly challenging and important area of machine learning.

All reviewers recognized the significance of the paper and voted for its acceptance. During the rebuttal phase, the authors addressed the reviewers' questions, added comparative baselines, and clarified the theoretical similarities between the scaling factors of low-rank and full-rank methods. Overall, I believe this is an excellent paper that should be published in NeurIPS. I therefore recommend its acceptance.